# KD-Zero: Evolving Knowledge Distiller for Any Teacher-Student Pairs

**Lujun Li**†* **Peijie Dong**† **Anggeng Li**† **Zimian Wei** **Ya Yang**
HKUST     HKUST(GZ)     Huawei     NUST     CityU
{lilujunai, dongpeijie98}@gmail.com,{anggeng.li,yya9}@outlook.com, weizimian16@nudt.edu.cn

## Abstract

Knowledge distillation (KD) has emerged as an effective technique for compressing models that can enhance the lightweight model. Conventional KD methods propose various designs to allow student model to imitate the teacher better. However, these handcrafted KD designs heavily rely on expert knowledge and may be sub-optimal for various teacher-student pairs. In this paper, we present a novel framework, KD-Zero, which utilizes evolutionary search to automatically discover promising distiller from scratch for any teacher-student architectures. Specifically, we first decompose the generalized distiller into knowledge transformations, distance functions, and loss weights. Then, we construct our distiller search space by selecting advanced operations for these three components. With sharpness and represent gap as fitting objectives, we evolve candidate populations and generate better distillers by crossover and mutation. To ensure efficient searching, we employ the loss-rejection protocol, search space shrinkage, and proxy settings during the search process. In this manner, the discovered distiller can address the capacity gap and cross-architecture challenges for any teacher-student pairs in the final distillation stage. Comprehensive experiments reveal that KD-Zero consistently outperforms other state-of-the-art methods across diverse architectures on classification, detection, and segmentation tasks. Noticeably, we provide some practical insights in designing the distiller by analyzing the distiller discovered. Codes are available in supplementary materials.

## 1 Introduction

Deep Neural Networks (DNNs) have achieved great success in tackling a variety of visual recognition tasks [31, 17, 66]. Despite the appealing performance, the prevailing DNN models usually have large numbers of parameters, leading to heavy costs of memory and computation. Conventional techniques such as Neural Architecture Search [16, 6, 26, 15] and quantizing networks to use low-bit parameters [11, 53, 49] have proven to be effective for mitigating this computational burden. Recently, Knowledge Distillation (KD) has been widely used for training compact and efficient neural networks by transferring the knowledge lied in the logits [23] or features [57] from a large, pre-trained teacher to a smaller student.

Recently, although KD has made significant progress in the hand-crafted designs, there are still some limitations to its practical

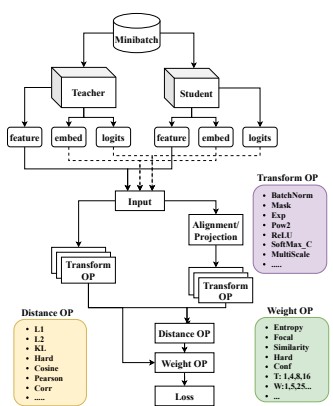

Figure 1: Illustration of search space in KD-Zero.

---

*Corresponding author, † Co-first authors, equal contribution.

37th Conference on Neural Information Processing Systems (NeurIPS 2023).

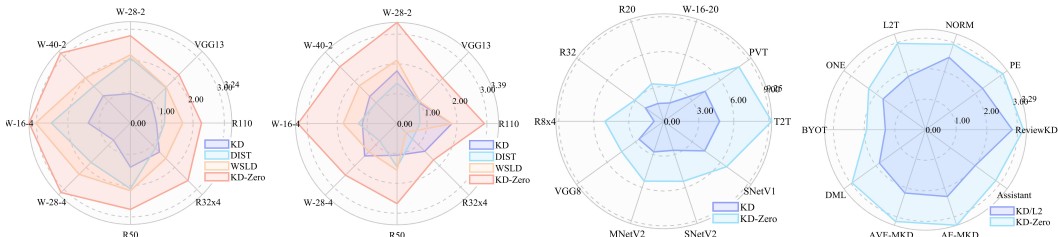

Figure 2: Top-1 gain (%). of WRN-16-2 (*left*) & ResNet-20 (*right*) distilled via various capacities teacher models on CIFAR-100.

Figure 3: Top-1 gain (%). of various student with KD-Zero (*left*). Various KD methods combined with KD-Zero (*right*) for ResNet-20 on C-100.

application for different scenarios from three aspects: **(1) Capacity gap problem:** Traditional KD methods do not always distill better from stronger teachers because of the large teacher-student gap [27, 46]. Larger and more accurate teacher models tend to be overconfident and fail to improve students [79]. To alleviate this, architecture-level methods use the assistant model [46] or architecture search [45], introducing additional training budgets. Other KD techniques use optimized designs or manually modified distillers to reduce the disparity. However, these techniques rely on expert knowledge and require careful tuning, which suffer from poor generality and efficiency. **(2) Cross-architecture issue.** Teacher-student networks with different architectural styles suffer from mismatches in position dependence, receptive fields, and feature dimensions. As shown in Figure 2, while handcrafted methods (*e.g.*, KD [23], DIST [27, 46] and WSLD [79]) provide slight gains in some cases, they still are weak in most cross-architecture pairs [2]. **(3) Manual tuning difficulties.** Different choices of knowledge sources, distillation functions, and hyperparameters can largely affect the performance of KD. However, designing and training a proper distillation setting requires trial and error, substantial effort, and experiments. Thus, two questions are raised: **(1) How to efficiently discover the optimal distillation strategies without expert knowledge? (2) How to reduce the teacher-student gap with different capabilities and architectures?**

*"We may hope that machines will eventually compete with men in all purely intellectual fields."*

— *Alan Turing*

As this well-said quote goes, recently, machine learning approaches have successfully replaced human experts in algorithm design, architecture search, and drug discovery. For the first problem, inspired by Auto-Zero [55], we decide to automate the process of designing the distillation functions for the first time in the field of knowledge distillation. For the second question, we systematically review previous studies of KD design for distillation gaps. Our observations reveal that: (1) Most of the distillers can be divided into three components with various key elements: knowledge transformation, distance function, and loss weights. (2) Some essential operations, like normalize ops [3] and mask ops [71], play important roles in reducing the teacher-student gap. (3) Optimal distillation functions can be effectively integrated with additional strategies (*e.g.*, feature aggregation [50], projector ensemble [9], N-to-One match [67], and cross-layer mapping [28, 13] in Figure 3).

Based on the above analysis, we present KD-zero, an automated search framework that utilizes evolutionary algorithms from scratch to efficiently discover the best distiller without manual design. Specifically, our framework is organized into three parts: search space, search algorithm, and acceleration strategy. Firstly, we establish the search space with basic transform operations, distance functions, and loss weights (see Figure 1). For example, we select operators normalized in different dimensions (*e.g.*, $batchnorm, norm_{HW,C,N}$), various types of activation functions (*e.g.*, exp, relu, tanh, sigmoid, pow2), multi-scale process and spatial-wise/channel-wise mask transforms and other advanced operations in the knowledge transformation. Our distance function options include smooth $\ell_1, \ell_1, \ell_2, \ell_{KL}, \ell_{hard}, \ell_{Cosine}, \ell_{Pearson}$ and $\ell_{Correlation}$ distance. Options in the loss weight part include various values for loss factors, temperature factors, and weight calibration strategies. In this way, our search space includes over 10,000 candidates covering the existing SOTA KD methods and designs. Then, we construct a calculation graph for these candidates and use selected features, representations, and logits from the teacher-student network as input. Based graph structure, we

---

[2]In our paper, W40-2, R32×4, R8×4, MV2, SV1, and SV2 stand for WRN-40-2, ResNet32×4, ResNet8×4, MobileNetV2, ShuffleNetV1, and ShuffleNetV2. DeiT-Ti [61] T2T-ViT-7 [72], PVT-Ti [64].

initialize the candidate total groups from the search space, crossover, and mutate them according to the evaluation results. We employ loss-rejection protocol and search space shrinkage for search efficiency to filter out weak candidates. With early-stop proxy settings, we achieve at least $40\times$ acceleration during the distiller search. For distillation gap reduction, we take the representation gap and sharpness gap between teacher-student as the fitting objectives besides the accuracy metric of the validation set. Finally, we distill student architectures with discovered distiller, and our KD-Zero surpasses existing KD approaches by a large margin without prior knowledge (see Figure 3).

In principle, our KD-Zero differs from previous hand-designed KD methods, opening new doors to automated distillation designs. Its merits can be highlighted in three aspects: (1) **Effective.** KD-Zero effectively reduces the teacher-student gap by presenting a general distiller search space and adaptive evolutionary search for different teacher-student pairs. KD-Zero extends the KD formulation and allows for additional gains with extra design besides distillers. In addition, it reduces human bias and ensures that the resulting distillers are optimized for the target problem or dataset. (2) **Efficient.** KD-Zero increases efficiency in practice by a series of flexible, systematic, and efficient search procedures without additional laborious tuning. By contrast, other manual methods with fixed KD forms involve complex parameter tuning with additional training time and resources. (3) **Insightful.** KD-Zero undertakes an in-depth analysis of the existing advanced distillation designs, with the aim of exploring their potential combination to produce numerous novel distillers. KD-Zero provides guidelines for practical applications and develops a new research direction. We hope our efforts on the automated design of distillers could facilitate future research for automated KD works to some extent. In summary, our contributions are:

- To alleviate architecture & capability gaps of teacher-student, we present KD-Zero, the first auto-search framework for evolving best distillers from scratch to our best knowledge.

- We present a comprehensive distiller search space, including advanced operations on transformations, distance functions, and loss weights. Then, we evolve the distiller search with performance and sharpness & represent the gap as fitting objectives. In addition, We achieve significant search acceleration via loss-rejection protocol & space shrinkage, and proxy settings.

- We conduct extensive experiments on classification, detection, and segmentation. KD-Zero performs state-of-the-art in multiple datasets and architectures (*e.g.*, CNN and vision transformer). Specifically, ResNet-18 and MobileNet with KD-Zero achieve 72.17% and 73.02% Top-1 accuracy on ImageNet, outperforming KD by 1.51%, 2.34%, respectively.

## 2 Related Work

**Knowledge Distillation.** The idea of teacher-student learning is first proposed in pioneering explorations [1, 2], and the formal definition is defined by the original KD [23]. Subsequent efforts explore on different knowledge (*e.g.*, intermediate feature representations [38, 43, 37, 33, 36], sample relationships [48, 60]) and applications [18, 14, 35]. **Compared to KD for distillation gap.** Previous methods propose assistant teachers, architecture search, KD designs on transformations [27], distance functions [59], and weight-tuning [42, 34] for this problem. However, such KD designs rely on expert knowledge and tuning, and their performance can fluctuate significantly across different situations. In contrast to these methods, KD-Zero develops automated searches for distillers to address these difficulties that do not require additional architecture modification and manual KD design. **Compared to Meta-KDs.** These works [13, 42] only focus on hyperparameter tuning and involve complex optimization challenges. In contrast, our approach searches for complementary distiller design besides hyperparameters establishing a new paradigm for KD research and application.

**Automated Machine Learning.** AutoML [81, 80] aims to automate Network Architecture Search (NAS) and HyperParameter Optimization (HPO), making them more accessible to non-experts. NAS chooses architecture rather than KD designs. **Compared to HPOs [54, 69],** they generally focus on the hyperparameters on training configurations. Recent methods search for loss formulation [39, 32]. In contrast to these methods, we present a new complex search space for distiller design in transforms, distances, loss weights, and new search objectives and accelerations according to the KD task.

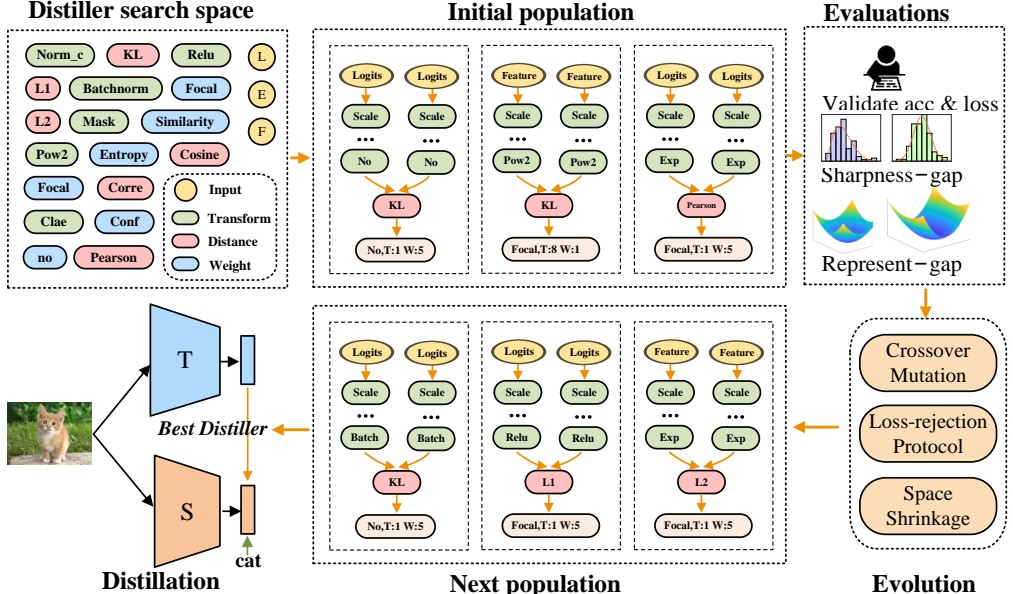

Figure 4: Overview of KD-Zero. In the search phase, we first randomly sample candidate distillers to initialize the population and evaluate their validation performance and sharpness & represent the gap between teacher-student. Then, we perform loss-rejection protocol & space shrinkage to remove weak individuals and crossover & mutation to generate new populations within promising ones. Finally, we pick up the best-performing distiller for distillation.

Table 1: Specific operations in KD-Zero. More details of their formulas are available in the Appendix.

| | |
|---|---|
| Transform | norm-based: $batchnorm, min-max, norm_{HW,C,N}, softmax_{HW,C,N}, logsoftmax_{HW,C,N}$ |
| | activation-based: $exp, mish, leaky, relu, tanh, sigmoid, pow2, pow4, log, sqrt$ |
| | scale-based: $scale, multi-scale, scale_{r1,r2}, local_{r1,r2,r4}, batch, channel$ |
| | attention&mask-based: $drop, satt, natt, catt, mask$, other: $no, bmm, mm$ |
| Distance | no-norm loss: smooth $\ell_1, \ell_1, \ell_2, \ell_{KL}, \ell_{hard}$; norm loss: $\ell_{Cosine}, \ell_{Pearson}, \ell_{Correlation}$ |
| Weight | calibration: $entropy, focal, sim, conf, no$; weight values: 0.01,..., 100; $\tau$ values:1,4, 8 |

## 3 Methodology

In this section, we first illustrate the design of our distiller search space, the search process, the acceleration, and the fitting objectives. Then we analyze the search results and give some guidelines. Finally, we analyze the student distilled via KD-Zero and expansion for different distillation scenarios. The pipeline of our approach is shown in Figure 4.

### 3.1 Search Space for Distillers Discovery

**Search space structure.** In KD, the student student $S$ is distilled with the fixed teacher $T$ by minimizing:

$$\mathcal{L}_{KD} = \tau^2 \times \mathcal{W}_f \times \mathcal{W}_{Cal} \times \mathcal{D}\big(\mathcal{T}(f_S/\tau), \mathcal{T}(f_T/\tau)\big), \tag{1}$$

where $\mathcal{W}_f$ and $\mathcal{W}_{Cal}$ is the loss weights factor and calibration [79], $\tau$ is the temperature factor, $\mathcal{T}$ is transformations, $\mathcal{D}(\cdot, \cdot)$ is distance function measuring the knowledge difference. $f_T$ and $f_S$ are outputs (*e.g.*, features, embeddings, and logits) of the teacher-student. Following this general KD formulation, our search space consists of different types of operations (see Table 1) in transformations, distance functions, and loss weights parts. Then, we use a computation graph to represent each candidate, in which the input nodes are different types of knowledge and the intermediate nodes are primitive operations. In addition, we assign three transform options as *transform-1→transform-2→transform-3* for the transformation part to ensure effective processing of the input knowledge.

**Insight of space design.** Our space design enjoys multiple merits. **(1) Comprehensive & flexible:** KD-Zero contains key elements of most existing KD methods, including the normalize, mask,

attention, and focal-calibration KDs [3, 79, 27] for distillation gaps. In addition, we introduce basic operations in loss design to improve the flexibility and diversity of the search. **(2) Extended & Innovative:** We extend advanced KD design for omni-dimensional and various knowledge. For example, we expand the normalized operation for batch, channel, and spatial dimensions, which can be used for logits, embeddings, and feature knowledge. Thus, such a unified and flexible search space would provide some inspiration for KD design.

### 3.2 Evolution Procedure, Acceleration and Objective

**Evolution process.** Our EA starts with an initial population of candidate distillers evaluated for their fitness based on our distillation gap-related multi-objectives. The algorithm then iteratively evolves the population over generations using genetic operators, such as selection, crossover, and mutation, to generate better solutions. Specifically, distillers are first randomly generated to form the initial population. Each candidate solution in the population is evaluated using multiple fitness functions that measure teacher-student gaps. Based on these evaluations, we select the best-performing individuals from the population to create a new population for the next generation. Then, we apply crossover to the selected individuals to create new offspring and use mutation to the new offspring to introduce diversity into the population, which helps to explore new distillers.

**Search acceleration** As the search space is sparse with many unpromising distillers, we employ several strategies to accelerate: **(1) Loss-rejection protocol.** We filter out candidates with excessive loss values or collapsed optimization during the search. **(2) Search space shrinkage.** We reduce sampling probabilities for the operations frequently in loss rejection and tail candidates with search iterations. **(3) Proxy settings.** With diverse and informative knowledge learned from a teacher, student models offer advantages in terms of faster training speeds. Based on these properties, we employ early stop the training process once the student model performs well enough to determine the quality of the candidate distillation. Nevertheless, proxy settings can also introduce evaluation uncertainties, and we alleviate this issue by introducing multiple distillation gap metrics in the following sections.

**Fitting objectives.** To accurately evaluate each distiller and reduce the distillation gap, we include cross-entropy loss, sharpness-gap [20] on prediction, and CKA-gap [52] on representation between teacher-student as the multi-objectives. Specifically, we conduct a training-free evolutionary search algorithm to efficiently discover the optimal distiller $\alpha^*$ from search space $\mathcal{A}$, as:

$$\alpha^* = \arg\min_{\alpha \in \mathcal{A}}(\mathcal{L}_{CE}(f_S, Y) + \overbrace{(log(\exp(f_S)) - log(\exp(f_T)))}^{Sharpness-gap} - \overbrace{\frac{\text{HSIC}(f_S, f_T)}{\sqrt{\text{HSIC}(f_S, f_S)\text{HSIC}(f_T, f_T)}}}^{CKA-gap}), \tag{2}$$

where $\mathcal{L}_{CE}$ is the regular cross-entropy objective with labels $Y$, the sharpness metric is the logarithm of the exponential sum of logits, and Centered Kernel Alignment (CKA) metric is normalized from Hilbert-Schmidt Independence Criterion (HSIC) [19] on high-level feature.

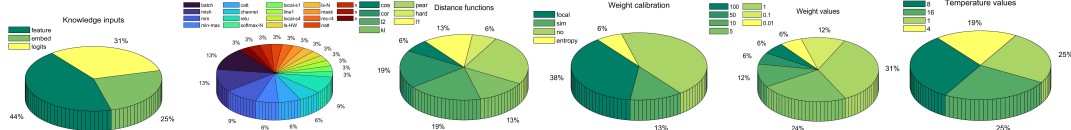

Figure 5: Probability of operations within each search part, which counted from the Top-3 searched distillers for all teacher-student pairs in the CIFAR-100 experiment.

### 3.3 Results Analysis and Practical Guidance

Figure 5 present searched distillers for different models. Based on these results, some practical guidance for KD designs can be summarized as follows:

- For knowledge input, feature knowledge enjoys superiority while logits and embedded knowledge share identical status. Searched distillers with feature knowledge take more proportion, and ablation studies on vanilla feature knowledge also surpass logits and embedding knowledge.

This observation also aligns with the existing conclusion that feature-based KD outperforms other KD on detection tasks [27, 63].

- For transformations, the normalized-based operations play a key role in most optimal distillers, and its vanilla performance outperforms other types of transformations in ablation studies. These observations illustrate that normalization benefits distillation, consistent with the current KD methods. Scale-based and activation-based operations are also crucial for the KD because they are often present in optimal distillers and enjoy good vanilla performance in ablation studies.

- For distance functions, normalized-based distances such as $\ell_{Pearson}$ and $\ell_{correlation}$ enjoy better results in ablation studies and are often adopted by teacher-student pairs across architectures. This suggests that these normalized-based distances can reduce distillation gaps in complex scenarios. In addition, some simple distance functions (*e.g.*, $\ell_1$, $\ell_2$ and $\ell_{KL}$) also appear in the optimal distiller. These may be attributed to these distillers employing advanced transformation operations for input knowledge.

- For loss weights, adjusting the temperature values is helpful for different scenarios, and the optimal weight values are generally between 1 and 10 based on search results and ablation studies. In addition, the focal weight calibration outperforms the no-weight calibration in ablation studies, and it is often used for some optimal distillers. In summary, we should employ smaller weight values and actively explore different weight calibration and temperature values to reduce distillation gaps under different teacher-student pairs.

### 3.4 Distilling Student via Discovered KD-Zero functions

After search, the discovered distiller $\mathcal{L}_{KD}$ is used for distilling student $f_S$ combined with cross-entropy $\mathcal{L}_{CE}$ via label $Y$ in single teacher $f_T$ or multiple teacher $f_{T_i}$ KD as:

$$\mathcal{L}_{single} = \mathcal{L}_{CE}(f_S, Y) + \mathcal{L}_{KD}(f_S, f_T), \quad \mathcal{L}_{multiple} = \mathcal{L}_{CE}(f_S, Y) + \sum_{i=1}^{N} \mathcal{L}_{KD}(f_S, f_{T_i'}). \quad (3)$$

**Extended to various distillation designs & scenarios** KD-Zero focuses on the distiller's search and employs simple $1\times1$-Conv for channel alignment. Recently, some KD methods have proposed other designs in feature aggregation [50], projector ensemble [9], N-to-One match [67], and cross-layer mapping [28, 13]. As shown in Table 2, KD-Zero can combine well with them by replacing their default $\ell_2/\ell_{KL}$ losses. In addition, the distiller search of KD-Zero also benefits different distillation scenarios. Specifically, Self-KDs, online KDs, and multi-teacher KDs with KD-Zero achieve extra gains than the default settings. Also, KD-Zero can combine architecture-level methods (*e.g.*, Assistant [46]) to reduce distillation gaps further.

Table 2: Top-1 (%) accuracy of KD-Zero combined with different KD designs and scenarios for ResNet-20 and WRN-16-2 on CIFAR-100.

| Net | Method | Different KD design | | | | Different KD scenarios | | | | | |
|---|---|---|---|---|---|---|---|---|---|---|---|
| | | Review [50] | PE [9] | NORM [67] | L2T-ww [28] | ONE [76] | BYOT [75] | DML [77] | AVE-MKD [58] | AE-MKD [58] | Assistant [46] |
| R20 | $\ell_{KL}$ or $\ell_2$ | 71.89 | 71.36 | 71.55 | 70.89 | 70.77 | 70.37 | 70.92 | 71.24 | 71.36 | 71.06 |
| | KD-Zero | 72.27 | 72.18 | 72.00 | 72.03 | 71.35 | 70.98 | 72.02 | 72.22 | 72.35 | 71.89 |
| W-16-2 | $\ell_{KL}$ or $\ell_2$ | 76.20 | 76.02 | 75.65 | 75.50 | 74.25 | 74.12 | 75.33 | 75.22 | 75.68 | 75.35 |
| | KD-Zero | 76.62 | 76.47 | 76.26 | 76.36 | 75.65 | 75.21 | 76.45 | 76.72 | 76.78 | 76.05 |

**Why can KD-Zero bridge the teacher-student gap?** The answer is intuitive: In KD-Zero, the search space contains many operators for distillation gap reduction, and search objectives are directly designed to reduce the prediction and representation gap between teacher-student. For example, We approximate the sharpness gap using a Taylor second expansion [20]:

$$G_{gap} = log(\exp(f_T)) - log(\exp(f_S)) \approx \log\left(1 + f_T + \frac{1}{2}f_T^2\right) - \log\left(1 + f_S + \frac{1}{2}f_S^2\right), \quad (4)$$

Following Hindon's assumption [23] that the logits of each training sample are approximately zero-meaned, i.e., $\bar{f}_T, \bar{f}_S = 0$. So the gap can be rewritten as $\log\left(1 + \frac{1}{2}Var(f_T)\right) - \log\left(1 + \frac{1}{2}Var(f_S)\right)$ [20]. Our options like normalized transform [3, 27], and weight calibrations [79] can effectively reduce the variance of teacher-studens' outputs and minimize the distillation gap. In addition, the student models distilled with KD-zero enjoy the merits: (1) Fast convergence and superior performance. As shown in Figure 6, KD-Zero has surpassed the best accuracies of KD

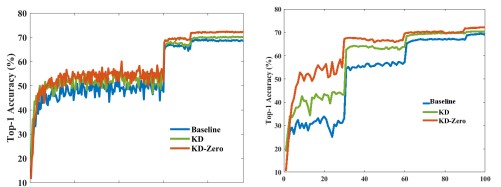
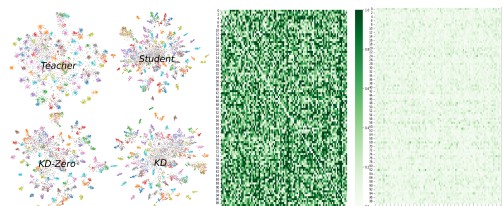

Figure 6: Comparison of training curves (*left*) of ResNet-20 on CIFAR-100 and ResNet-18 on ImageNet (*right*).

Figure 7: Penultimate-layer visualization (*left*), logits-correlation map of teacher-student (ResNet-110/20) via KD (*middle*) & KD-Zero (*right*).

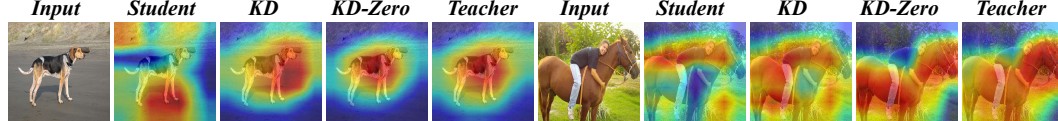

Figure 8: Comparison on the Grad-CAM++ [4] visualization results between the features of the baseline student model, the student model trained with KD and our KD-Zero, and the teacher model. Results are obtained on ImageNet with ResNet18 (left) and MobileNet (right).

and baseline at the beginning of the 3rd learning-rate decay stage. (2) Smaller teacher-student gap. As shown in Figure 7, features and logits of students via KD-Zero have stronger similarities with teachers than the original KD method.

## 4 Experiments

In this section, we assess the efficacy of our proposed KD-Zero approach on classification, detection, and segmentation tasks. Additionally, we compare its performance with other KD methods, ensuring fair comparisons by utilizing the same training settings. We report mean results based on more than 3 repeated trials. More detailed experiment settings and results are available in the Appendix.

Table 3: Comparison of results on CIFAR-100. Most results of other methods refer to the original papers [8, 60]. *Gain* refers to the performance gain than baseline. We report Top-1 "mean (std)" accuracies (%) for KD-Zero over 3 runs.

| Model | Same architectural style | | | | | Different architectural style | | | | | |
|---|---|---|---|---|---|---|---|---|---|---|---|
| Teacher | W-40-2 | R110 | R110 | R32×4 | VGG13 | VGG13 | R32×4 | W-40-2 | R56 | R56 | R56 |
| Student | W-16-2 | R20 | R32 | R8×4 | VGG8 | MNetV2 | SNetV2 | SNetV1 | DeiT | T2T | PVT |
| Teacher | 75.61 | 74.31 | 74.31 | 79.42 | 74.64 | 74.64 | 79.42 | 75.61 | 72.34 | 72.34 | 72.34 |
| Student | 73.26 | 69.06 | 71.14 | 72.50 | 70.36 | 64.60 | 71.82 | 70.50 | 65.08 | 69.37 | 69.22 |
| FitNets [57] | 73.58 | 68.99 | 71.06 | 73.50 | 71.02 | 64.14 | 73.54 | 73.73 | 70.82 | 71.96 | 73.42 |
| AT [73] | 74.08 | 70.22 | 72.31 | 73.44 | 71.43 | 59.40 | 72.73 | 73.32 | 73.51 | 74.01 | 73.60 |
| SP [62] | 73.83 | 70.04 | 72.69 | 72.94 | 72.68 | 66.30 | 74.56 | 74.52 | 67.36 | 72.26 | 70.48 |
| RKD [48] | 73.35 | 69.25 | 71.82 | 71.90 | 71.48 | 64.52 | 73.21 | 72.21 | 70.39 | 71.88 | 73.63 |
| CRD [60] | 75.48 | 71.46 | 73.48 | 75.51 | 73.94 | 69.73 | 75.65 | 76.05 | NA | NA | NA |
| SRRL [29] | 75.46 | 71.51 | 73.80 | 75.92 | 73.23 | 69.34 | 75.66 | 76.61 | NA | NA | NA |
| KD [23] | 74.92 | 70.67 | 73.08 | 73.33 | 72.98 | 67.37 | 74.45 | 74.83 | 73.25 | 74.15 | 73.68 |
| DIST [27] | 75.35 | 71.68 | 73.86 | 75.79 | 73.86 | 69.17 | 76.08 | 75.85 | 72.56 | 73.86 | 73.52 |
| WSLD [79] | 75.30 | 71.53 | 73.36 | 74.79 | 73.86 | 68.79 | 75.93 | 75.09 | 74.56 | 75.28 | 74.39 |
| IPWD [47] | NA | 71.32 | 73.91 | 76.03 | NA | NA | NA | 76.03 | NA | NA | NA |
| **KD-Zero** | **76.42** | **72.05** | **74.19** | **77.85** | **75.26** | **70.42** | **77.45** | **77.52** | **78.25** | **78.32** | **77.22** |
| Gain$_{\pm STD}$ | 3.16$_{\pm0.16}$ | 2.99$_{\pm0.21}$ | 3.05$_{\pm0.12}$ | 5.35$_{\pm0.22}$ | 4.90$_{\pm0.11}$ | 5.82$_{\pm0.18}$ | 5.63$_{\pm0.25}$ | 7.02$_{\pm0.14}$ | 13.17$_{\pm0.36}$ | 8.95$_{\pm0.29}$ | 8.00$_{\pm0.32}$ |

### 4.1 Experiments on CIFAR-100

**Implementation**. We utilize the CIFAR-100 dataset [30] in knowledge distillation. During the distiller search phase, we apply 5% early-stopping training epochs with full training data for acceleration settings. Our evolutionary algorithm with 20 population sizes performs 100 iterations for each teacher-student pair. During the distillation phase, all teacher-student networks are trained using typical training settings, with a training epoch of 240. The multi-step learning rate commences at 0.1, which decays by 0.1 at 100 and 150 epochs. Recently, knowledge distillation enabled training Vision Transformers (ViT) from scratch with CNNs as teachers. To evaluate the effectiveness of KD-Zero, we conduct the evolutionary search for ViT-based distillation strategies with the same settings as the

Table 4: Comparison of results on ImageNet. The results of other methods quote the original papers report [8, 60, 71]. We report Top-1 "mean (std)" accuracies (%) for KD-Zero.

| Teacher | Student | Acc. | Teacher | Student | KD [23] | AT [73] | OFD [22] | SRRL [29] | CRD [60] | Review [50] | MGD [71] | **KD-Zero** |
|---------|---------|------|---------|---------|---------|---------|----------|-----------|----------|-------------|----------|-------------|
| ResNet-34 | ResNet-18 | Top-1 | 73.40 | 69.75 | 70.66 | 70.69 | 70.81 | 71.73 | 71.17 | 71.61 | 71.58 | **$72.17_{\pm 0.15}$** |
|  |  | Top-5 | 91.42 | 89.07 | 89.88 | 90.01 | 89.98 | 90.60 | 90.13 | 90.51 | 90.35 | **$90.46_{\pm 0.25}$** |
| ResNet-50 | MobileNet | Top-1 | 76.16 | 70.13 | 70.68 | 70.72 | 71.25 | 72.49 | 71.37 | 72.56 | 72.35 | **$73.02_{\pm 0.22}$** |
|  |  | Top-5 | 92.86 | 89.49 | 90.30 | 90.03 | 90.34 | 90.92 | 90.41 | 91.00 | 90.71 | **$91.05_{\pm 0.26}$** |

CNN experiment. Subsequently, we train the ViT with the optimal distiller obtained and ResNet-56 as CNN teacher. The training is conducted on $224 \times 224$ resolution images for 300 epochs, with an initial learning rate of 5e-4 and a weight decay 0.05 using the AdamW optimizer.

**Comparison results on CNN models.** Table 3 presents a comparative analysis of our KD-Zero with other state-of-the-art (SOTA) KD methods. We conduct multiple trials with randomly selected distillers in the same search space, called Rand-KD, to evaluate the efficacy of our EA search. For teacher-student pairs with the same architectural style, KD-Zero outperforms the baselines by margins ranging from $3.16\% \sim 4.90\%$. Compared with Rand-KD and other KDs, KD-Zero obtains consistent performance gains ($1.0\% \sim 3.2\%$). Besides strengths in the same architecture pairs, KD-Zero exhibits even stronger performance when dealing with different architectural styles, while other KD methods suffer from noticeable accuracy reductions. Specifically, KD-Zero outperforms the baseline by margins of $5.6\% \sim 7.3\%$ and the random search results by margins of $1.9\% \sim 2.3\%$, demonstrating the effectiveness of our design for different structures. Compared with other SOTA KD methods, our KD-Zero achieves $1.2\% \sim 1.5\%$ gains. These results show that KD-Zero can improve each student model with simple settings under different teacher-student pairs.

**Comparison results on vision transformer.** Table 3 presents the results of the vanilla and distillation models employing different distillation methods. The results indicate that KD-Zero can significantly improve the performance of vision transformers with $8.0\% \sim 13.1\%$ margins and consistently yields superior performance than other methods. In addition, our proposed method applies to various ViT architectures, thereby validating its effectiveness. Note that most ViT students possess larger model sizes (*e.g.*, DeiT-Ti with 5 million parameters) and greater capabilities than the CNN teacher (*e.g.*, ResNet-56 with only 0.86 million parameters). Some ViT students outperform the CNN model in the strong regularization setting on large-scale datasets. However, when it comes to ViT distillation on small datasets, employing CNN teachers helps address the issue of ViT models struggling to train effectively from scratch. In this context, using CNN teachers in distillation is akin to auxiliary training or providing additional regularization supervision. As a result, these ViTs demonstrate their original strong representation capability after distillation and consequently outperform the CNN teacher.

## 4.2   Experiments on ImageNet

**Implementation**. We additionally conduct experiments on the ImageNet[12]. Following CIFAR-100 trials, we employ similar EA settings on a subset of ImageNet for search acceleration. Then, we utilize the discovered distiller for the training of student models (*e.g.*, ResNet-18 [21] and MobileNet [25]). The training settings are the same as the other KD methods and involve training for 100 epochs using a multi-step learning rate, which commences at 0.1 and decays by 0.1 at 30, 60, and 90 epochs.

**Comparison results.** As shown in Table 4, our proposed KD-Zero significantly improves the accuracy of baseline models, yielding gains of $2.5\% \sim 2.9\%$ in Top-1 accuracy for ResNet-18 and MobileNet, respectively. In addition, KD-Zero surpasses other SOTA methods with clear gains, demonstrating its superiority in large-scale datasets. These findings substantiate the effectiveness of KD-Zero in distillation optimization with considerable benefits, establishing the versatility and potency of our framework. In summary, KD-Zero facilitates substantially improved predictive accuracy of student models on ImageNet, more complex domains while preserving superior performance.

**Visualizations.** The comparison between the Grad-CAM++ maps generated by the student model trained with KD-Zero and other methods is presented in Figure 8. The results indicate that the Grad-CAM++ map generated by the student model trained with KD-Zero is more similar to that of the teacher model compared to the student model trained independently. In contrast, independent training of the student model leads to incorrect focus areas. These findings suggest that the KD-Zero

Table 5: **The results on the COCO val dataset for teacher (T) and student (S) models.**

| Method | AP | $AP_{50}$ | $AP_{75}$ | $AP_S$ | $AP_M$ | $AP_L$ |
|---|---|---|---|---|---|---|
| *Two-stage detectors* | | | | | | |
| T: Cascade Mask RCNN-X101 | 45.6 | 64.1 | 49.7 | 26.2 | 49.6 | 60.0 |
| S: Faster RCNN-R50 | 38.4 | 59.0 | 42.0 | 21.5 | 42.1 | 50.3 |
| KD [24] | 39.7 | 61.2 | 43.0 | 23.2 | 43.3 | 51.7 |
| FKD [74] | 41.5 | 62.2 | 45.1 | 23.5 | 45.0 | 55.3 |
| CWD [59] | 41.7 | 62.0 | 45.5 | 23.3 | 45.5 | 55.5 |
| DIST [27] | 40.4 | 61.7 | 43.8 | 23.9 | 44.6 | 52.6 |
| KD-Zero | 41.9 | 62.7 | 45.5 | 23.6 | 45.6 | 55.6 |
| *One-stage detectors* | | | | | | |
| T: RetinaNet-X101 | 41.0 | 60.9 | 44.0 | 23.9 | 45.2 | 54.0 |
| S: RetinaNet-R50 | 37.4 | 56.7 | 39.6 | 20.0 | 40.7 | 49.7 |
| KD [24]* | 37.2 | 56.5 | 39.3 | 20.4 | 40.4 | 49.5 |
| FKD [74] | 39.6 | 58.8 | 42.1 | 22.7 | 43.3 | 52.5 |
| FGD [70] | 40.4 | 59.9 | 43.3 | 23.4 | 44.7 | 54.1 |
| DIST [27] | 39.8 | 59.5 | 42.5 | 22.0 | 43.7 | 53.0 |
| KD-Zero | 40.9 | 60.4 | 43.5 | 23.2 | 45.2 | 54.8 |

Table 6: **Results on Cityscapes val dataset with ImageNet Pretrain.**

| Method | mIoU (%) |
|---|---|
| T: DeepLabV3-R101 | 78.07 |
| S: DeepLabV3-R18 | 74.21 |
| SKD [44] | 75.42 |
| IFVD [65] | 75.59 |
| CWD [59] | 75.55 |
| CIRKD [68] | 76.38 |
| DIST [27] | 77.10 |
| KD-Zero | 77.38 |
| S: PSPNet-R18 | 72.55 |
| SKD [44] | 73.29 |
| IFVD [65] | 73.71 |
| CWD [59] | 74.36 |
| CIRKD [68] | 74.73 |
| KD-Zero | 76.25 |

Figure 9: Ablation study of search space (*left*) and organization (*right*) of ResNet-20 on C-100.

Figure 10: Comparison of search algorithms (*left*) and correlation visualization (*right*) of ResNet-20.

approach is more effective than traditional knowledge distillation methods in guiding the student model to learn from the teacher model, resulting in improved model interpretability and performance.

### 4.3 Experiments on Object Detection & Semantic Segmentation

**Object detection.** We conduct experiments on the MS-COCO dataset[41]. We use the optimal distiller on ImageNet to distill knowledge from teacher detectors to students. Based on the strong baseline [5], we apply KD-Zero to two-stage detector (*e.g.*, Faster R-CNN [56]) and the one-stage detector (*e.g.*, RetinaNet [40]), which are widely used object detection frameworks. Following common practice [40], all models are trained with a $2\times$ learning schedule (24 epochs). We train all the models with SGD optimizer, where the momentum is 0.9, and the weight decay is 0.0001. As shown in Table 5, our KD-Zero improves the AP by 3.5 on RetinaNet and 3.4 on Faster R-CNN, respectively, outperforms previous state-of-the-art techniques, including [59, 70, 74], for both object detectors. The results substantiate the potential of KD-Zero for scaling knowledge transfer to broader datasets and more complex computer vision problems while preserving improved accuracy.

**Semantic segmentation.** We evaluate KD-Zero on Cityscapes dataset[10]. Following the previous work, we adopt PSPNet-ResNet101 [78, 51] as the teacher and PSPNet and DeepLabV3[7] models with the ResNet18 backbone as the student. During distillation, the batch size is 8, and the models are trained for 40K iterations with the SGD optimizer, where the momentum is 0.9 and the weight decay is 0.0005. The results are reported with mean Intersection-over-Union (mIoU) under the single-scale evaluation setting. As shown in Table 6, the student PSPNet and DeepLabV3 get 3.17 and 3.7 mIoU improvement by adding our KD-Zero loss. These results indicate that our method surpasses the state-of-the-art distillation method for semantic segmentation, demonstrating that searched distillers facilitate student learning.

Table 7: Top-1 (%) accuracy of ResNet20 (baseline : 69.06%) with different teachers models on CIFAR-100. Note that the distiller is searched with ResNet110 as the teacher.

| Teacher | R110 | VGG13 | W-28-2 | W-40-2 | W-16-4 | W-28-4 | R50 | R32×4 |
|---|---|---|---|---|---|---|---|---|
|  | 74.31 | 74.64 | 75.45 | 75.61 | 77.51 | 78.60 | 79.34 | 79.42 |
| KD | $70.89_{\pm0.13}$ | $70.07_{\pm0.23}$ | $70.83_{\pm0.21}$ | $70.35_{\pm0.14}$ | $70.22_{\pm0.13}$ | $70.59_{\pm0.11}$ | $70.12_{\pm0.08}$ | $70.37_{\pm0.11}$ |
| DIST | $69.82_{\pm0.16}$ | $70.10_{\pm0.21}$ | $70.42_{\pm0.13}$ | $70.05_{\pm0.21}$ | $70.35_{\pm0.21}$ | $69.88_{\pm0.21}$ | $70.51_{\pm0.21}$ | $69.79_{\pm0.21}$ |
| WSLD | $70.89_{\pm0.17}$ | $70.12_{\pm0.18}$ | $71.19_{\pm0.23}$ | $70.85_{\pm0.26}$ | $70.86_{\pm0.22}$ | $70.39_{\pm0.16}$ | $70.62_{\pm0.22}$ | $69.48_{\pm0.24}$ |
| KD-Zero | $72.05_{\pm0.21}$ | $71.21_{\pm0.12}$ | $72.46_{\pm0.08}$ | $71.76_{\pm0.16}$ | $72.37_{\pm0.11}$ | $71.55_{\pm0.11}$ | $71.74_{\pm0.17}$ | $70.98_{\pm0.14}$ |

Table 8: Top-1 (%) accuracy of WRN-16-2 (baseline : 73.26%) with different teachers models on CIFAR-100. Note that the distiller is searched with WRN-40-2 as the teacher.

| Teacher | R110 | VGG13 | W-28-2 | W-40-2 | W-16-4 | W-28-4 | R50 | R32×4 |
|---|---|---|---|---|---|---|---|---|
|  | 74.31 | 74.64 | 75.45 | 75.61 | 77.51 | 78.60 | 79.34 | 79.42 |
| KD | $74.09_{\pm0.23}$ | $74.21_{\pm0.24}$ | $74.20_{\pm0.26}$ | $74.49_{\pm0.18}$ | $74.61_{\pm0.14}$ | $74.03_{\pm0.12}$ | $74.67_{\pm0.18}$ | $74.57_{\pm0.12}$ |
| DIST | $74.35_{\pm0.11}$ | $74.86_{\pm0.13}$ | $75.34_{\pm0.22}$ | $74.92_{\pm0.15}$ | $75.79_{\pm0.17}$ | $75.04_{\pm0.18}$ | $75.36_{\pm0.22}$ | $74.39_{\pm0.25}$ |
| WSLD | $74.93_{\pm0.28}$ | $74.88_{\pm0.19}$ | $75.44_{\pm0.13}$ | $75.30_{\pm0.16}$ | $76.29_{\pm0.26}$ | $75.58_{\pm0.18}$ | $75.42_{\pm0.21}$ | $74.86_{\pm0.26}$ |
| KD-Zero | $75.50_{\pm0.24}$ | $75.45_{\pm0.15}$ | $76.05_{\pm0.21}$ | $76.42_{\pm0.16}$ | $76.50_{\pm0.19}$ | $76.40_{\pm0.18}$ | $76.02_{\pm0.16}$ | $75.86_{\pm0.15}$ |

## 4.4 Ablation Studies

**Search space organization.** We individually explore various operation types and organized settings in Figure 9. Results indicate: (1) some operations like the normalized-based transforms, normalized-based losses, focal correction, and small-weighted values bring superior gains than others. (2) our design is optimal across different settings, and the removal of transform, weight factor, and temperature factor search leads to worse results, proving our organization's strength.

**Search algorithm.** We use the EA for distiller search, which is gradient-free and flexible for non-convex optimization. As shown in Figure 10, the EA obtains faster convergence and better final search results than random search. In addition, for fitting objectives in EA, Shapeness and CKA metrics demonstrate good correlations with the final distillation accuracy and combine well with cross-entropy loss to improve evaluation quality.

**Generalization of searched distillers for different Teachers.** In Table 8 & 7, we present detailed results of the same searched distiller with various capacities teacher models on CIFAR-100. The results indicate that our KD-Zero can significantly improve the student model in different teacher models and surpass the KD and improved KD methods (*e.g.*, DIST [27], WSLD [79]).

In addition, we provide combined trials in Table 2 and Figure 5, student training and teacher-student gap analysis in Figure 6 & 7. These ablation studies demonstrate that our KD-Zero can well alleviate distillation gaps and combine with other KD designs.

## 5 Conclusion

In this paper, we introduce the KD-Zero framework, an innovative framework for automatically designing distillers. We construct a flexible and unified distiller search space with advanced operators on knowledge transformation, distance function, and loss weight parts. Then, we employ evolutionary search with distillation-gap reduction as objectives and search acceleration strategies, enabling KD-Zero to efficiently find distillers and improve student model performance in distillation training. We employ the loss-rejection protocol, search space shrinkage, and proxy settings during the search process to enhance search efficiency. Furthermore, we present some practical guidelines based on the results analysis and propose various extensions of KD-Zero to cater to different KD designs and scenarios. Comprehensive experiments on three benchmarks demonstrate the effectiveness and universality of the KD-Zero framework for various CNNs, Vision Transformer models, object detection, and semantic segmentation. In future work, we will extend KD-Zero to more tasks and explore large language models as code encoders or searcher for KD-Zero. We hope this work will inspire future research on knowledge distillation designs.

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
