# Supplementary Materials for *KD-Zero: Evolving Knowledge Distiller for Any Teacher-Student Pairs*

**Lujun Li**†*  **Peijie Dong**†  **Anggeng Li**†  **Zimian Wei**  **Ya Yang**
HKUST      HKUST(GZ)      Huawei      NUST      CityU
{lilujunai, dongpeijie98}@gmail.com,{anggeng.li,yya9}@outlook.com, weizimian16@nudt.edu.cn

## A   Preliminary Designs for KD-GPT.

Large-scale language model like GPT has a strong ability to generate code text and encode knowledge information. As demonstrated in Algorithm 1, we present a preliminary design of KD-GPT, a GPT-guided distiller research search framework to leverage the capabilities of large-scale language models to accelerate the knowledge distillation search process and discover better distillers. Based on the original KD-Zero research search process, GPT can be used to identify potential variant candidates with its encoding of practical laws corresponding to good distillers. Furthermore, the KD-GPT method enables the automatic generation of complete distiller codes based on user-provided options. This innovative approach builds upon the original KD-Zero research search process, integrating GPT models to enhance the efficiency and accuracy of knowledge distillation. By exploiting the code generation and knowledge encoding potential of GPT, KD-GPT paves the way for more effective and streamlined distillation processes, ultimately improving the performance of machine learning models.

---

**Algorithm 1** Evolution Search for KD-GPT

---

**Input**: Search space $\mathcal{S}$, population $\mathcal{P}$, max iteration $\mathcal{T}$, sample ratio $r$, sampled pool $\mathcal{R}$, topk $k$.
**Output**: Distiller with highest score.
 1: $\mathcal{P}0 :=$ Initialize population($P_i$);
 2: Sample pool $\mathcal{R} := \emptyset$;
 3: **for** $i = 1, 2, \ldots, \mathcal{T}$ **do**
 4:     Clear sample pool $\mathcal{R} := \emptyset$;
 5:     Randomly select $\mathcal{R} \in \mathcal{P}$;
 6:     Candidates $G_i k :=$ GetTopk($\mathcal{R}$, $k$);
 7:     Parent $G_i^p :=$ RandomSelect($G_{i_k}$);
 8:     Mutate $G_i^m :=$ MUTATE($G_i^p$);
 9:     **// GPT-Guided Selection.**
10:     Randomly initialize $E$;
11:     **if** $\rho(G_i^m) > \rho(E)$ **then**
12:        Append $G_i^m$ to $P$;
13:     **else**
14:        Append $E$ to $P$;
15:     **end if**
16:     Remove the distiller with the lowest score.
17: **end for**

---

## B   Limitations.

Following most general KDs [4, 14, 17, 12, 16, 13, 15, 18] and AutoML methods [6, 10, 1, 5, 43, 42], the limitation of current our automatic search method for knowledge distillation strategies is that they are typically evaluated on classification tasks and assume that the distilled knowledge will transfer

---

*Corresponding author, † Co-first authors, equal contribution.

37th Conference on Neural Information Processing Systems (NeurIPS 2023).

well to downstream tasks. However, this may not always be the case, as different downstream tasks may require different types of knowledge to be distilled. To address this limitation, future work could expand the search space to include task-specific design choices for downstream tasks, such as object detection and semantic segmentation. By incorporating task-specific constraints and objectives into the search process, it may be possible to identify more effective and efficient knowledge distillation strategies that are tailored to the specific requirements of different applications. Additionally, it will be essential to evaluate the effectiveness of these methods on a range of downstream tasks to ensure that they generalize well and can be widely applied in real-world scenarios.

## C  Broader Impacts.

The proposed method aims to automatically distill design knowledge from human engineers and design experts and represent it in a formal and computational model. This has the potential to advance the field of machine learning and artificial intelligence, particularly in machine design automation. By capturing and codifying human design expertise, we can develop machine learning models to serve as automated design assistants and co-designers. This can enable technological advances and productivity gains in engineering design processes. No broad social or ethical implications exist since this is focused on technical neural architecture search methods. The impact would be limited to potential performance gains for machine learning or AI applications that use neural networks, with no significant effects outside of the technical machine learning research community. The ability to find high-performing neural architectures more efficiently through training-free search techniques could provide benefits by enabling faster progress and iteration in neural architecture research. However, it is essential to note that this concept is primarily of interest to researchers in the field of machine learning and may not have immediate or direct implications for broader society or ethics.

Table 1: Correlation coefficients between validation metrics under proxy settings and actual distillation performance of ResNet-20 on CIFAR-100, which is calculated from 10 different candidates. S-gap means Sharpness-gap.

| CE Loss | S-gap | CKA-gap | CE Loss+S-gap | CE Loss+CKA-gap | CE Loss+S-gap+CKA-gap |
|---------|-------|---------|---------------|-----------------|------------------------|
| 0.755   | 0.512 | 0.682   | 0.824         | 0.885           | 0.935                  |

Table 2: Search time and of KD-Zero with different hyperparameters for evolutionary search configurations. The search time is measured on a single GeForce RTX™ 3090 Ti GPU.

| Hyperparam | Iterations | | | Population size | | |
|------------|-----|------|------|-----|------|------|
|            | 50  | 100  | 300  | 10  | 20   | 30   |
| Search time | 10h | 21h | 63h | 11h | 21h | 30h |

## D  More Ablation Studies and Results

### D.1  Ablation Studies about Search Settings

In Table 1, we also examine the correlation between our validation metrics and the actual accuracy for 10 random candidates in Table 1. These results indicate that both our loss-rejection protocol and search space shrinkage contribute to improving search results and that sharpness-gap and CKA-gap are also more accurate in predicting the best distillers.

### D.2  Sensitivity Analysis of Hyperparameters

Our evolutionary search introduces hyperparameters about the iteration numbers and population size to balance the search cost and results. As shown in Table 2, (20, 100) is the best option for iteration and population size, which obtains the preferable trade-off on efficiency and accuracy. Less iterations and population size would introduce a significant loss of accuracy. While larger numbers settings increase search time significantly but improve accuracy slightly.

### D.3 Analysis of Search Costs

In Table 2, we present the search time of KD-Zero. Like the sensitivity analysis for hyperparameters, our current search settings are well-balanced in efficiency and accuracy. In future work, we will explore more acceleration strategies, such as parallel computation, to reduce search costs.

### D.4 Analysis of Training Costs

We compare the training speed of our KD-Zero with vanilla KD [8], RKD [24], CRD [31], and DML [39], as summarized in Table 3. Our KD-Zero has almost the same highest training speed as the vanilla KD, outperforming other KD methods.

Table 3: Average training time consumption (batch time) on CIFAR-100, which measured in a single GeForce RTX™ 3090 Ti GPU.

| Student | KD | PKT | CRD | RKD | DML | KD-Zero |
|---|---|---|---|---|---|---|
| R20 | 0.219 | 0.225 | 0.234 | 0.234 | 0.524 | 0.222 |
| R32 | 0.319 | 0.336 | 0.353 | 0.846 | 0.912 | 0.328 |
| VGG8 | 0.097 | 0.120 | 0.139 | 0.207 | 0.190 | 0.105 |

## E  Detailed Analysis of Distiller Search Space

In this section, we provide a detailed formulation and discussion of distillation operations in our search space. Their detailed implementations are presented in Listings 1-4.

### E.1 Knowledge Transformations

**Attention transformation.** Unlike matching the original features [29], the attention distillation method applies feature distillation on the attention maps. For example, AT [37] summarizes the values across the channel dimension and transfers the attention knowledge to the student. FKD [38] uses the sum of teacher and student attention to focus the student on changeable areas. FGD [35] proposes focal distillation, which forces the student to learn the teacher's crucial parts, and global distillation, which compensates for missing global information.

Following FGD [35], we select the attention transformation operation on different pixels and different channels, respectively:

$$G^S(F) = \frac{1}{C} \cdot \sum_{c=1}^{C} |F_c|, A^S(F) = H \cdot W \cdot \sigma\big(G^S(F)/\tau\big), \quad (1)$$

$$G^C(F) = \frac{1}{HW} \cdot \sum_{i=1}^{H} \sum_{j=1}^{W} |F_{i,j}|, A^C(F) = C \cdot \sigma\big(G^C(F)/\tau\big), \quad (2)$$

where $H$, $W$, $C$ denote the height, width, and channel of the feature. $G^S$ and $G^C$ are the spatial and channel attention maps. $A^S$ and $A^C$ are the spatial and channel attention masks, where $\tau$ is the temperature hyperparameter to adjust the distribution.

**Mask transformation.** Other distillation methods address the gap between the teacher and student models by performing the mask transformation operation before distilling the features. Following MGD [36], we use the $l$-th mask to cover the $l$-th feature of the student. This can be expressed as:

$$M_{i,j}^l = \begin{cases} 0, & \text{if } R_{i,j}^l < \lambda \\ 1, & \text{Otherwise} \end{cases} \quad (3)$$

where $R_{i,j}^l$ is a random number between 0 and 1, and $i$ and $j$ are the horizontal and vertical coordinates of the feature map, respectively.

**Multi-scale transformation.** Multi-scale feature representations are beneficial for capturing context information at different levels of abstraction and are essential for many vision tasks. For instance,

PSPNet [40] uses spatial pyramid pooling to extract convolutional features at multiple scales for semantic image segmentation. Following KR [2], our multi-scale transformation operation extracts different levels of knowledge from the feature by using spatial pyramid pooling.

**Local transformation.** Local features, such as person Re-ID and image retrieval, are commonly used in computer vision tasks. Local features refer to distinctive and repeatable patterns or structures within an image that can be used to identify and match corresponding features in different images. LKD [19] uses a local correlation matrix based on selected local parts to guide the student's learning. In our search space, we select the local transformation operation to divide the original feature into $n^2$ patches (e.g., $n = 2, 4$), and then distill each patch separately.

**Sample-wise transformation.** The relationships between input samples are complex and make it difficult for teachers to transfer knowledge to students. For example, RKD [25] involves comparing the similarity of angle and structure distances between samples. CC [26] is another sample-wise knowledge distillation method that focuses on capturing the correlations between embedding representations. Following SP [33], we use the correlation matrix for the sample-wise features in the following way:

$$\mathcal{L}_{KD} = \left\| \frac{(\tilde{A}^T) \cdot (\tilde{A}^T)^\top}{\left\| (\tilde{A}^T) \cdot (\tilde{A}^{T\,\top}) \right\|_2} - \frac{(\tilde{A}^S) \cdot (\tilde{A}^S)^\top}{\left\| (\tilde{A}^S) \cdot (\tilde{A}^{S\,\top}) \right\|_2} \right\|_2, \tag{4}$$

where $\tilde{A}^T, \tilde{A}^S \in \mathbf{R}^{N \times CHW}$ are the reshaped versions of the original features $A^T$ and $A^S$. Additionally, we minimize the Kullback-Leibler (KL) divergence for the sample relationships with the following equation:

$$\mathcal{L}_{KD} = \frac{\mathcal{T}^2}{N} \sum_{n=1}^{N} \sum_{i=1}^{C \cdot W \cdot H} \phi\left(\tilde{A}_{n,i}^T\right) \cdot \log\left[\frac{\phi\left(\tilde{A}_{n,i}^T\right)}{\phi\left(\tilde{A}_{n,i}^S\right)}\right] \tag{5}$$

where $\phi$ is the softmax function and $\mathcal{T}$ is the temperature coefficient. Additionally, sample distillation is combined with $L_2$ distance when the distilling feature is normalized in the sample dimension.

**Channel-wise transformation.** Recent distillation methods also focus on the knowledge contained in each channel. These methods aim to improve the matching of channel-wise features by optimizing or generating attention-weighting factors. For example, CWD [30] minimizes the KL divergence between the probability map, which is calculated by normalizing the feature map of each channel. ICKD [22] calculates the disparity of the channel correlation matrix. Following ICKD [22], channel-wise features $G^S$ and $G^T$ are transformed using channel-wise operations and then computed using the $L_2$ loss as follows:

$$\mathcal{L}_{KD} = \left\| \frac{(G^T) \cdot (G^T)^\top}{\|(G^T) \cdot (G^T)^\top\|_2} - \frac{(G^S) \cdot (G^S)^\top}{\|(G^S) \cdot (G^S)^\top\|_2} \right\|_2, \tag{6}$$

Additionally, the channel-wise knowledge can also be derived from the following equation:

$$\mathcal{L}_{KD} = \frac{\mathcal{T}^2}{C} \sum_{c=1}^{C} \sum_{i=1}^{W \cdot H} \phi\left(G_{c,i}^T\right) \cdot \log\left[\frac{\phi\left(G_{c,i}^T\right)}{\phi\left(G_{c,i}^S\right)}\right] \tag{7}$$

where $\phi$ is the softmax function and $\mathcal{T}$ is the temperature coefficient. Note that channel distillation and $L_2$ distance are combined when normalization is achieved in the channel dimension.

### E.2 Distance Functions

Different distance functions are used to measure the difference between the output of the teacher and student networks in the process of distillation. Let $P_i$ denote the predicted probability of class $i$ by the teacher network, and $Q_i$ denote the predicted probability of class $i$ by the student network.

$L_2$ **distance**: The $L_2$ distance measures the square root of the sum of the squared differences between the probabilities of each class in the two distributions. The $L_2$ distance between $P$ and $Q$ is defined as:

$$D_{L_2}(P,Q) = \sqrt{\sum_{i=1}^{n}(P_i - Q_i)^2}$$

**Cosine distance**: The cosine distance measures the cosine of the angle between the two probability vectors. This distance measure is useful when the magnitudes of the probability vectors are not important, only their directions. The cosine distance between $P$ and $Q$ is defined as:

$$D_{Cosine}(P,Q) = 1 - \frac{\sum_{i=1}^{n} P_i Q_i}{\sqrt{\sum_{i=1}^{n} P_i^2}\sqrt{\sum_{i=1}^{n} Q_i^2}}$$

When normalization is applied to both distributions, i.e., $\sum_{i=1}^{n} P_i = \sum_{i=1}^{n} Q_i = 1$, the cosine distance is equivalent to the $L_2$ distance as follows:

$$D_{L_2,norm} = \sqrt{\frac{1}{n^2}\sum_{i=1}^{n}(P_i - Q_i)^2} \tag{8}$$

$$= \sqrt{\frac{1}{n^2}\sum_{i=1}^{n} P_i^2 - \frac{2}{n^2}\sum_{i=1}^{n} P_i Q_i + \frac{1}{n^2}\sum_{i=1}^{n} Q_i^2} \tag{9}$$

$$= \sqrt{\frac{2}{n^2}\sum_{i=1}^{n}(P_i^2 + Q_i^2 - P_i Q_i)} \tag{10}$$

$$= \sqrt{2 \times \left(1 - D_{Cosine}(P,Q)\right)} \tag{11}$$

**Pearson distance**: The Pearson distance measures the correlation between the two probability vectors. The Pearson distance between $P$ and $Q$ is defined as:

$$D_{Pearson}(P,Q) = 1 - \frac{\sum_{i=1}^{n}(P_i - \bar{P})(Q_i - \bar{Q})}{\sqrt{\sum_{i=1}^{n}(P_i - \bar{P})^2}\sqrt{\sum_{i=1}^{n}(Q_i - \bar{Q})^2}}$$

where $\bar{P}$ and $\bar{Q}$ are the means of the two distributions. Similarly, the Pearson distance is also correlated with the normalized $L_2$ distance.

**KL distance**: The KL distance measures the information lost when approximating the probability distribution $P$ with the probability distribution $Q$, as follows:

$$D_{KL}(P,Q) = \sum_{i=1}^{n} P_i \log \frac{P_i}{Q_i} = \sum_{i=1}^{n} P_i \log P_i - \sum_{i=1}^{n} P_i \log Q_i$$

**Correlation distance**: Let $P \in^{n \times d}$ and $Q \in^{n \times d}$ denote a batch of representations from the student and teacher networks, respectively. These matrices are computed before the final fully-connected layer to preserve the structural information of the data, thus enabling a strong distillation signal for the student. We first normalize these representations to have zero mean and unit variance across the batch dimension and then propose to construct a cross-correlation matrix, $\mathbf{C}_{st} = P^T Q/n \in^{d \times d}$. A perfect correlation between the two sets of representations is achieved if all of the diagonal entries $v_i = (\mathbf{C}_{st})_{ii}$ are equal to one. To formulate this as a minimization problem, we propose the following loss:

$$D_{corr}(P,Q) = \log_2 \sum_{i=1}^{d} |v_i - 1|^{2\alpha} \tag{12}$$

### E.3 Loss Weights

**Temperature factor**: The temperature coefficient $\mathcal{T}$ is used to scale the divergence between the probability distributions of the teacher and student networks. When $\mathcal{T}$ is small, the probability distribution of the teacher network becomes sharper, and the probability distribution of the student network is forced to be closer to the teacher's distribution. On the other hand, when the temperature coefficient $\mathcal{T}$ is large, the probability distribution of the teacher network becomes more diffuse, and the student network is allowed to produce more varied and less precise probability estimates. In our search space, we set 1, 4, 8, and 16 as options for the temperature coefficient.

**Loss weights factor**: Loss weights play an important role in determining the behavior of the distillation process. Adjusting the weights makes it possible to balance different optimization objectives and tailor the student model's behavior to the task's specific requirements. In our search space, we set 100, 50, 25, 10, 5, 1, 0.1, and 0.01 as options for the loss weights.

**Loss weights calibration**: In light of recent knowledge distillation works that assign sample-wise weights, we propose 'entropy,' 'focal,' 'sim,' 'conf,' and 'no' as options for loss weights calibration in the knowledge distillation loss. In particular, 'entropy' refers to the use of the entropy of the teacher's knowledge as a weighting factor, 'conf' is the confidence level of the teacher's model knowledge, and 'sim' refers to the teacher-student similarity. Following WSLD [41], 'focal' calibration is assigned to each sample's loss according to the predictions of the teacher and the student, which is formally defined as

$$L_{\text{focal}} = \left( 1 - \exp \left( -\frac{\log \hat{y}_{i,1}^s}{\log \hat{y}_{i,1}^t} \right) \right) L_{\text{kd}} = \left( 1 - \exp \left( -\frac{L_{\text{ce}}^s}{L_{\text{ce}}^t} \right) \right) L_{\text{kd}}, \tag{13}$$

where $i$ is the ground truth class of the sample.

# F   Detailed Experimental Settings

This section provides detailed settings for the experiments conducted on the CIFAR-100 and ImageNet datasets. All experiments follow standard training settings without any additional data augmentation or other training techniques.

## F.1   Experiments on CIFAR-100

**Dataset**. CIFAR-100 [11] is a popular classification dataset used to evaluate the performance of distillation methods. It consists of 50,000 training images and 10,000 test images, with 100 classes.

**Implementation**. In the comparison experiments with other offline KD methods, we use the same training settings as CRD [32] to implement various KD methods. The training epochs for all methods are set to 240. We use a mini-batch size of 64 and a standard SGD optimizer with a weight decay of $5 \times 10^{-4}$. The multi-step learning rate is initialized to 0.05 and decayed by 0.1 at 150, 180, and 210 epochs. To compare with existing knowledge distillation methods, we refer to the original settings and CRD to implement various knowledge methods. The configuration details, including the factor ($\lambda$), are the same as those used in the CIFAR-100 experiments.

## F.2   Experiments on ImageNet

**Dataset**. We also conduct experiments on the ImageNet dataset (ILSVRC12) [3], which is considered the most challenging classification task. It consists of approximately 1.2 million training images and 50 thousand validation images, belonging to 1,000 categories.

**Implementation**. In the ImageNet experiments, the student models (ResNet-18 [7] and MobileNet [9]) are trained for 100 epochs. The batch size is set to 256, and the multi-step learning rate is initialized to 0.1, with decay at 30, 60, and 90 epochs. Other KD methods are implemented following the hyperparameter settings in the original paper. Moreover, the detailed settings for KD-Zero are the same as those used in the CIFAR-100 experiments.

## F.3   Experiments on Vision Transformer

**Implementation**: Recently, Knowledge Distillation (KD) has emerged as an efficient method to train ViTs [34] with Convolutional Neural Networks (CNNs) as teachers. In this study, we aim to evaluate the effectiveness of KD-Zero by searching for ViT-based distillation strategies on the CIFAR-100 dataset. To ensure fair comparison, we use the same data augmentation and regularization methods as described in DeiT, including Auto-Augment, Rand-Augment, and mixup. We conduct a distiller search using the same settings as in the CNN experiment. Once we find the optimal distiller, we train the ViT model using ResNet-56 as the CNN teacher. The training process is performed on images with a resolution of $224 \times 224$ for 300 epochs. We use an initial learning rate of 5e-4, a weight decay of 0.05, and the AdamW optimizer. The batch size is set to 128, and the learning rate schedule follows the cosine policy.

## F.4   Experiments on Object Detection.

**Dataset**. We evaluate KD-Zero on the MS-COCO dataset [21], which contains over 120K images covering 80 categories. All performance is evaluated on the MS-COCO validation set.

**Implementation**. We apply KD-Zero to two-stage detectors (e.g., Faster R-CNN [28]) and one-stage detectors (e.g., RetinaNet [20]), which are widely used object detection frameworks. We initialize the backbone with weights pre-trained on ImageNet [3]. Following common practice [20], all models are trained with a $2\times$ learning schedule (24 epochs). Horizontal image flipping is utilized in data augmentation. For KD-Zero, we build an extra shadow head with the same architecture as the original classification head, which performs distillation in the fine-tuning detector stage.

# G    Experiments on Semantic Segmentation

**Datasets.** Cityscapes is a challenging benchmark that has been collected from 50 cities in order to gain a better understanding of urban scenes. It consists of 5,000 carefully annotated images belonging to 19 different classes. The images are divided into three sets: 2,975 for training, 500 for validation, and 1,525 for testing. Additionally, there are 20,000 images with less detailed annotations, which were used in the knowledge distillation experiments [27].

**Implementation details:** In this section, we use the mean Intersection-over-Union (mIoU) as the evaluation metric for all experiments. The results are reported in the single-scale evaluation setting. We start by using DeeplabV3 with a ResNet101 backbone as the teacher model, as done in prior works [23]. For the other distillation methods, we explore DeepLabV3 models with a ResNet18 backbone. For KD-Zero, we choose DeepLabV3 with students searched by KD-Zero on ImageNet. During the distillation process, we use SGD as the optimizer with a poly-learning-rate policy. Each training image is randomly cropped into 512x512 pixels. The batch size is 8, and the models are trained for 40K iterations unless specified otherwise.

Listing 1: The PyTorch implementation of unified KD-Zero loss.

```python
import torch
import torch.nn as nn
from torch import Tensor
import torch.nn.functional as F
from einops import rearrange, reduce, repeat
from einops.layers.torch import Rearrange, Reduce
'''
trans_func1, trans_func2, trans_func3:
'exp','mish','leaky','relu','tanh','sigmoid','pow2','pow4','log','sqrt',
'drop','no', 'satt','natt','catt','mask','bmm','mm','batch','channel',
'scale_r1','scale_r2','multi_scale_r4','local_s1','local_s2','local_s4',
'norm_HW','norm_C','norm_N','softmax_N','softmax_C','softmax_HW','scale',\\
'logsoftmax_N','logsoftmax_C','logsoftmax_HW','min_max_normalize','batchnorm'\\
'''
class UnifiedLoss(nn.Module):
   def __init__(self,
      knowledge_input=params['knowledge_input'], #'feat', "emd",'logit'
      trans_func1=params['trans_func1'],
      trans_func2=params['trans_func2'],
      trans_func3=params['trans_func3'],
      distance_func=params['distance_func'], #
          'l1','l2','kl','hard','smooth_l1','cos','pear','cor'
      weight=params['weight'], # 'entropy','focal','sim','conf','no'
      temperature=params['kd_T'], # 1, 4, 8, 16
      magnitude=params['gamma'], # 100, 50, 25, 10, 5, 1, 0.1, 0.01
      **kwargs,
   ):
      super().__init__()
      self.Knowledge_input = knowledge_input
      self.trans_func1 = trans_func1
      self.trans_func2 = trans_func2
      self.trans_func3 = trans_func3
      self.distance_func = distance_func
      self.weight = weight
      self.temperature = temperature
      self.magnitude = magnitude
   def forward(self, feat_s, feat_t, emb_s, emb_t, logit_s, logit_t,
      gt_label):
      if self.Knowledge_input == 'feat':
         teacher, student = feat_t, feat_s
```

```python
        elif self.Knowledge_input == 'emb':
            teacher, student = emb_t, emb_s
        else:
            teacher, student = logit_t, logit_s
        trans_func1 = _TRANS_FUNC1[self.trans_func1]
        trans_func2 = _TRANS_FUNC2[self.trans_func2]
        trans_func3 = _TRANS_FUNC3[self.trans_func3]
        distance_func = _DIS_FUNC[self.distance_func]
        weight = _WIT_FUNC[self.weight]
        f_t =
            trans_func3(trans_func2(trans_func1(student)))/self.temperature
        f_s =
            trans_func3(trans_func2(trans_func1(teacher)))/self.temperature
        return self.magnitude * self.temperature * self.temperature *
            weight(distance_func(f_s, f_t), student, teacher, gt_label)
```

Listing 2: The PyTorch implementation of transformation operations.

```python
import torch
import torch.nn as nn
from torch import Tensor
import torch.nn.functional as F
from einops import rearrange, reduce, repeat
from einops.layers.torch import Rearrange, Reduce

def trans_norm_HW(f):
    """transform with l2 norm in HW dim"""
    if len(f.shape) == 2:
        return f
    elif len(f.shape) == 3:
        return F.normalize(f, p=2, dim=2)
    elif len(f.shape) == 4:
        return F.normalize(f, p=2, dim=(2, 3))
    else:
        raise f'invalide shape {f.shape}'

def trans_norm_C(f):
    """transform with l2 norm in C dim"""
    return F.normalize(f, p=2, dim=1)

def trans_norm_N(f):
    """ transform with l2 norm in N dim"""
    return F.normalize(f, p=2, dim=0)

def trans_multi_scale_r1(f):
    """transform with multi-scale distillation with reduce ratio of 1"""
    if len(f.shape) != 4:
        return f
    return reduce(f, 'b c (h1 h2) (w1 w2) -> b c h1 w1', 'max', h2=1,
        w2=1)

def trans_multi_scale_r2(f):
    """transform with multi-scale distillation with reduce ratio of 2"""
    if len(f.shape) != 4:
        return f
    return reduce(f, 'b c (h1 h2) (w1 w2) -> b c h1 w1', 'max', h2=2,
        w2=2)

def trans_multi_scale_r4(f):
    """transform with multi-scale distillation with reduce ratio of 4"""
    if len(f.shape) != 4:
        return f
    return reduce(f, 'b c (h1 h2) (w1 w2) -> b c h1 w1', 'max', h2=4,
        w2=4)
```

```python
def trans_local_s1(f):
    """transform with local features distillation with spatial size of
        1"""
    if len(f.shape) != 4:
        return f
    f = rearrange(f, 'b c (h hp) (w wp) -> b (c h w) hp wp', hp=1, wp=1)
    return f.squeeze(-1).squeeze(-1)

def trans_local_s2(f):
    """transform with local features distillation with spatial size of
        1"""
    if len(f.shape) != 4:
        return f
    return rearrange(f, 'b c (h hp) (w wp) -> b (c h w) hp wp', hp=2,
        wp=2)

def trans_local_s4(f):
    """transform with local features distillation with spatial size of
        1"""
    if len(f.shape) != 4:
        return f
    return rearrange(f, 'b c (h hp) (w wp) -> b (c h w) hp wp', hp=4,
        wp=4)

def trans_batch(f):
    """transform with batch-wise shape"""
    if len(f.shape) == 2:
        return f
    elif len(f.shape) == 3:
        return rearrange(f, 'b c h -> b (c h)')
    elif len(f.shape) == 4:
        return rearrange(f, 'b c h w -> b (c h w)')

def trans_channel(f):
    """transform with channel-wise shape"""
    if len(f.shape) in {2, 3}:
        return f
    elif len(f.shape) == 4:
        return rearrange(f, 'b c h w -> b c (h w)')

def trans_mask(f, threshold=0.65):
    """transform with mask"""
    if len(f.shape) in {2, 3}:
        # logits
        return f
    N, C, H, W = f.shape
    device = f.device
    mat = torch.rand((N, 1, H, W)).to(device)
    mat = torch.where(mat > 1 - threshold, 0, 1).to(device)
    return torch.mul(f, mat)

def trans_satt(f, T=0.5):
    """transform with spatial attention"""
    if len(f.shape) in {2, 3}:
        # logits
        return f
    N, C, H, W = f.shape
    value = torch.abs(f)
    fea_map = value.mean(axis=1, keepdim=True)
    # Bs*W*H
    S_attention = (H * W * F.softmax(
        (fea_map / T).view(N, -1), dim=1)).view(N, H, W)
    return S_attention.unsqueeze(dim=-1)
```

```python
def trans_natt(f, T=0.5):
    """transform from the N dim"""
    if len(f.shape) == 2:
        N, C = f.shape
    elif len(f.shape) == 4:
        N, C, H, W = f.shape
    elif len(f.shape) == 3:
        N, C, M = f.shape
    # apply softmax to N dim
    return N * F.softmax(f / T, dim=0)

def trans_catt(f, T=0.5):
    """transform with channel attention"""
    if len(f.shape) == 2:
        # logits
        N, C = f.shape
        # apply softmax to C dim
        return C * F.softmax(f / T, dim=1)
    elif len(f.shape) == 3:
        N, C, M = f.shape
        return C * F.softmax(f / T, dim=1)
    elif len(f.shape) == 4:
        N, C, H, W = f.shape
        value = torch.abs(f)
        # Bs*C
        channel_map = value.mean(
            axis=2, keepdim=False).mean(
                axis=2, keepdim=False)
        C_attention = C * F.softmax(channel_map / T, dim=1)
        return C_attention.unsqueeze(dim=-1).unsqueeze(dim=-1)
    else:
        raise f'invalid shape {f.shape}'

def trans_drop(f, p=0.1):
    """transform with dropout"""
    return F.dropout2d(f, p)

def trans_nop(f):
    """no operation transform """
    return f

def trans_bmm(f):
    """transform with gram matrix -> b, c, c"""
    if len(f.shape) == 2:
        return f
    elif len(f.shape) == 4:
        return torch.bmm(
            rearrange(f, 'b c h w -> b c (h w)'),
            rearrange(f, 'b c h w -> b (h w) c'))
    elif len(f.shape) == 3:
        return torch.bmm(
            rearrange(f, 'b c m -> b c m'), rearrange(f, 'b c m -> b m c'))
    else:
        raise f'invalide shape {f.shape}'

def trans_mm(f):
    """transform with gram matrix -> b, b"""
    if len(f.shape) == 2:
        return f
    elif len(f.shape) == 3:
        return torch.mm(
            rearrange(f, 'b c m -> b (c m)'), rearrange(f, 'b c m -> (c m)
                b'))
    elif len(f.shape) == 4:
```

```python
        return torch.mm(
            rearrange(f, 'b c h w -> b (c h w)'),
            rearrange(f, 'b c h w -> (c h w) b'))
    else:
        raise f'invalide shape {f.shape}'

def trans_norm_HW(f):
    """transform with l2 norm in HW dim"""
    if len(f.shape) == 2:
        return f
    elif len(f.shape) == 3:
        return F.normalize(f, p=2, dim=2)
    elif len(f.shape) == 4:
        return F.normalize(f, p=2, dim=(2, 3))
    else:
        raise f'invalide shape {f.shape}'

def trans_norm_C(f):
    """transform with l2 norm in C dim"""
    return F.normalize(f, p=2, dim=1)

def trans_norm_N(f):
    """ transform with l2 norm in N dim"""
    return F.normalize(f, p=2, dim=0)

def trans_softmax_N(f):
    """transform with softmax in 0 dim"""
    return F.softmax(f, dim=0)

def trans_softmax_C(f):
    """transform with softmax in 1 dim"""
    return F.softmax(f, dim=1)

def trans_softmax_HW(f):
    """transform with softmax in 2,3 dim"""
    if len(f.shape) == 2:
        return f
    if len(f.shape) == 4:
        N, C, H, W = f.shape
        f = f.reshape(N, C, -1)
    assert len(f.shape) == 3
    return F.softmax(f, dim=2)

def trans_logsoftmax_N(f):
    """transform with logsoftmax"""
    return F.log_softmax(f, dim=1)

def trans_logsoftmax_C(f):
    """transform with logsoftmax"""
    return F.log_softmax(f, dim=1)

def trans_logsoftmax_HW(f):
    """transform with logsoftmax"""
    if len(f.shape) == 2:
        return f
    if len(f.shape) == 4:
        N, C, H, W = f.shape
        f = f.reshape(N, C, -1)
    assert len(f.shape) == 3
    return F.log_softmax(f, dim=2)

def trans_sqrt(f):
    """transform with sqrt"""
    return torch.sqrt(f)
```

```python
def trans_log(f):
    """transform with log"""
    return torch.sign(f) * torch.log(torch.abs(f) + 1e-9)

def trans_pow2(f):
    """transform with ^2"""
    return torch.pow(f, 2)

def trans_pow4(f):
    """transform with ^4"""
    return torch.pow(f, 4)

def trans_min_max_normalize(f):
    """transform with min-max normalize"""
    A_min, A_max = f.min(), f.max()
    return (f - A_min) / (A_max - A_min + 1e-9)

def trans_abs(f):
    """transform with abs"""
    return torch.abs(f)

def trans_sigmoid(f):
    """transform with sigmoid"""
    return torch.sigmoid(f)

def trans_swish(f):
    """transform with swish"""
    return f * torch.sigmoid(f)

def trans_tanh(f):
    """transform with tanh"""
    return torch.tanh(f)

def trans_relu(f):
    """transform with relu"""
    return F.relu(f)

def trans_leaky_relu(f):
    """transform with leaky relu"""
    return F.leaky_relu(f)

def trans_mish(f):
    """transform with mish"""
    return f * torch.tanh(F.softplus(f))

def trans_exp(f):
    """transform with exp"""
    return torch.exp(f)

def trans_scale(f):
    """transform 0-1"""
    return (f + 1.0) / 2.0

def trans_batchnorm(f):
    """transform with batchnorm"""
    if len(f.shape) in {2, 3}:
        bn = nn.BatchNorm1d(f.shape[1]).to(f.device)
    elif len(f.shape) == 4:
        bn = nn.BatchNorm2d(f.shape[1]).to(f.device)
    return bn(f)
```

Listing 3: The PyTorch implementation of distance function operations.

```python
import torch
```

```python
import torch.nn as nn
from torch import Tensor
import torch.nn.functional as F
from einops import rearrange, reduce, repeat
from einops.layers.torch import Rearrange, Reduce

def l1_loss(f_s: Tensor, f_t: Tensor) -> Tensor:
    """l1_loss = (f_s - f_t).abs()"""
    return F.l1_loss(f_s, f_t, reduction='none')

def l2_loss(f_s: Tensor, f_t: Tensor) -> Tensor:
    """mse_loss = l2_loss = (f_s - f_t) ** 2"""
    return F.mse_loss(f_s, f_t, reduction='none')

def kl_loss(f_s: Tensor, f_t: Tensor) -> Tensor:
    """kl_loss = kl_divergence = f_s * log(f_s / f_t)"""
    return F.kl_div(f_s, f_t, reduction='none')

def hard_loss(f_s: Tensor, f_t: Tensor) -> Tensor:
    """kl_loss = kl_divergence = f_s * log(f_s / f_t)"""
    return F.cross_entropy(f_s, f_t.argmax(dim=1), reduction='none')

def smooth_l1_loss(f_s: Tensor, f_t: Tensor) -> Tensor:
    """smooth_l1_loss = (f_s - f_t).abs()"""
    return F.smooth_l1_loss(f_s, f_t, reduction='none')

def cosine_similarity(f_s, f_t, eps=1e-8):
    """cosine_similarity = f_s * f_t / (|f_s| * |f_t|)"""
    return F.cosine_similarity(f_s, f_t, eps=eps)

def pearson_correlation(f_s, f_t, eps=1e-8):
    """pearson_correlation = (f_s - mean(f_s)) * (f_t - mean(f_t)) /
        (|f_s - mean(f_s)| * |f_t - mean(f_t)|)"""

    def cosine(f_s, f_t, eps=1e-8):
        return (f_s * f_t).sum(1) / (f_s.norm(dim=1) * f_t.norm(dim=1) +
            eps)

    return 1 - cosine(f_s - f_s.mean(1).unsqueeze(1),
                    f_t - f_t.mean(1).unsqueeze(1), eps)

def correlation(z_s, z_t):
    f_s = z_s
    f_t = z_t
    n, d = f_s.shape
    f_s_norm = (f_s - f_s.mean(0)) / f_s.std(0)
    f_t_norm = (f_t - f_t.mean(0)) / f_t.std(0)
    c_st = torch.einsum('bx,bx->x', f_s_norm, f_t_norm) / n
    c_diff = c_st - torch.ones_like(c_st)
    alpha = 1.01
    c_diff = torch.abs(c_diff)
    c_diff = c_diff.pow(2.0)
    c_diff = c_diff.pow(alpha)
    return torch.log2(c_diff.sum())
```

Listing 4: The PyTorch implementation of loss weight operations.

```python
import torch
import torch.nn as nn
from torch import Tensor
import torch.nn.functional as F
from einops import rearrange, reduce, repeat
from einops.layers.torch import Rearrange, Reduce
```

```python
def teacher_entropy(f: Tensor, t: Tensor = None, s: Tensor = None,
    gt_label: Tensor = None):
    if t is None and s is None:
        return f * tensor(100).to(f.device)
    if len(t.shape) == 1:
        N = t.shape
    elif len(t.shape) == 2:
        N, C = t.shape
    elif len(t.shape) == 3:
        N, C, M = t.shape
    elif len(t.shape) == 4:
        N, C, H, W = t.shape
    else:
        raise f'invalid shape {t.shape}'
    tea_std = torch.std(t, dim=-1,keepdim=True)
    entropy_weight = tea_std
    print(entropy_weight)
    return (entropy_weight * entropy_weight * f).mean()

def teacher_student_similarity(f: Tensor, t: Tensor = None, s: Tensor =
    None, gt_label: Tensor = None):
    if t is None and s is None:
        return f * tensor(100).to(f.device)
    if len(t.shape) == 1:
        N = t.shape
    elif len(t.shape) == 2:
        N, C = t.shape
    elif len(t.shape) == 3:
        N, C, M = t.shape
    elif len(t.shape) == 4:
        N, C, H, W = t.shape
    else:
        raise f'invalid shape {t.shape}'
    # compute cosine similarity for t and s
    similarity = F.cosine_similarity(s, t)
    return (similarity * f).mean()

def focal(f: Tensor, logit_s: Tensor = None, logit_t: Tensor = None,
    gt_label: Tensor = None):
    s_input_for_softmax = logit_s
    t_input_for_softmax = logit_t
    t_soft_label = F.softmax(t_input_for_softmax, dim=1)
    logsoftmax = nn.LogSoftmax()
    softmax_loss = - torch.sum(t_soft_label *
        logsoftmax(s_input_for_softmax), 1, keepdim=True)
    fc_s_auto = logit_s.detach()
    fc_t_auto = logit_t.detach()
    log_softmax_s = logsoftmax(fc_s_auto)
    log_softmax_t = logsoftmax(fc_t_auto)
    one_hot_label = F.one_hot(gt_label, num_classes=100).float()
    softmax_loss_s = - torch.sum(one_hot_label * log_softmax_s, 1,
        keepdim=True)
    softmax_loss_t = - torch.sum(one_hot_label * log_softmax_t, 1,
        keepdim=True)
    focal_weight = softmax_loss_s / (softmax_loss_t + 1e-7)
    ratio_lower = torch.zeros(1).cuda()
    focal_weight = torch.max(focal_weight, ratio_lower)
    focal_weight = 1 - torch.exp(- focal_weight)
    return (focal_weight * f).mean()

def confidence(f: Tensor, logit_s: Tensor = None, logit_t: Tensor =
    None, gt_label: Tensor = None):
    s_input_for_softmax = logit_s
```

```
    t_input_for_softmax = logit_t
    t_soft_label = F.softmax(t_input_for_softmax, dim=1)
    logsoftmax = nn.LogSoftmax()
    softmax_loss = - torch.sum(t_soft_label *
        logsoftmax(s_input_for_softmax), 1, keepdim=True)
    fc_s_auto = logit_s.detach()
    fc_t_auto = logit_t.detach()
    log_softmax_s = logsoftmax(fc_s_auto)
    log_softmax_t = logsoftmax(fc_t_auto)
    one_hot_label = F.one_hot(gt_label, num_classes=100).float()
    softmax_loss_s = - torch.sum(one_hot_label * log_softmax_s, 1,
        keepdim=True)
    softmax_loss_t = - torch.sum(one_hot_label * log_softmax_t, 1,
        keepdim=True)
    confidence_weight = 1.0 - softmax_loss_s
    return (confidence_weight * f).mean()

def no_weight(f: Tensor, logit_s: Tensor = None, logit_t: Tensor = None,
    gt_label: Tensor = None):
    return f.mean()
```