# OpenReview forum: "KD-Zero: Evolving Knowledge Distiller for Any Teacher-Student Pairs"
_NeurIPS.cc/2023/Conference — NeurIPS 2023 poster_

### Official Review · Reviewer_WY3i · 2023-07-01

**Soundness:** 3 good
**Presentation:** 3 good
**Contribution:** 3 good
**Rating:** 7
**Confidence:** 4

**Summary:**

Knowledge distillation has received significant attention over the last decade and numerous KD approaches have been proposed so far. This paper focuses on automatically learning the best distillation strategy/loss for a given dataset and student/teacher pair. The distiller search space is composed of various transformation ops, distance functions, loss factors, temperature factors and loss calibration functions. The best distiller in this broad search space is searched using an evolutionary search process. To make the search process efficient, strategies such as loss-rejection protocol and search space shrinkage are used. Experiments were conducted on several datasets and the discovered distillation strategy (which varies based on the dataset and teacher-student combinations) is shown to perform better than various existing distillation strategies.

**Strengths:**

This paper brings AuotML into research area of knowledge distillation and shows that by automatic searching we can find better distillation strategies than many existing KD approaches. I think this is interesting and useful for the community to know.

Thorough experimental evaluation: Experiments were conducted on many datasets (recognition, detection, segmentation) and teacher-student network pairs.



**Weaknesses:**

Some results are confusing: In Table 4, the Auto-KD performance of transformer-based students (last three columns) is significantly better than the teacher model (5-6%). Makes me wonder if the student was finetuned from some pretrained checkpoint instead of training from scratch.

Limited ablations: The proposed search process uses three criteria: test loss, sharpness-gap and CKA-gap. What is the contribution of sharpness-gap and CKA-gap? Ablation studies on this are very limited. Results with only one teacher-student combination on CIFAR100 are presented in the supplementary material (Table 3).  Based on these results, the CKA-gap and sharpness-gap criteria do not seem to be that important (the performance without these is close to the standard deviation range of the performance when they are used).

**Questions:**

Please see weaknesses section.

---

> ### Author Rebuttal · Authors · 2023-08-07
>
> Dear Reviewer WY3i,
>
> Thank you for your constructive feedback and insightful questions. We appreciate your time and effort in reviewing our work. We would like to address your concerns as follows:
>
> ----
>
> **Q1:** About some results in Table 4.
>
> **A1:** The students in our experiments are indeed trained from scratch and not fine-tuned from pre-trained checkpoints. Some  ViT distilled by CNN teachers achieves superior results. The reasons are as follows:
>
> 1. ViT relies on self-attention mechanisms to capture global dependencies among image patches, allowing it to effectively model long-range interactions. Such ViT architecture performs well with large-scale datasets and strong regularization. However, small datasets are not sufficient to extract hierarchical information for ViT. This leads to weak results for these ViT models vanilla trained from scratch on CIFAR-100.
>
> 2. CNN teacher models are good at extracting local and hierarchical feature knowledge, which ViT models exactly need. With the auxiliary supervision from the CNN teacher, the ViT model can train well on CIFAR-100 and present strong performance.
>
> 3. we provide additional results on ViT-T on large-scale ImageNet datasets in the following Table. Our KD-Zero can improve the ViT from  72.0 to 76.0, surpassing the traditional KD methods e.g., DeiT by large margins, demonstrating our method's generality.
>
>    | Teacher      | Student | Teacher | Student | KD    | DeiT | DearKD-Ti | KD-Zero |
>    | ------------ | ------- | ------- | ------- | ----- | ---- | --------- | ------- |
>    | RegNetY-16GF | ViT-T   | 82.9    | 72.2    | 72.35 | 74.5 | 74.8      | 76.0    |
>
> **Q2:** About sharpness-gap and CKA-gap.
>
> **A2:** **1. Contributions:** The sharpness-gap and CKA-gap are important criteria for evaluating the distiller's effectiveness in our proxy settings, determining the search accuracy-efficiency trade-offs. In addition, the criteria help distillers bridge the gap between teachers and students in the search by minimizing them as objectives. The training curves,  T-SNE visualization, logits-correlation in Figures 6 & 7, and Grad-CAM++  visualization in Supp. Figure 1 illustrate that selecting distillers with these criteria results in smaller gaps between student and teacher models.
>
> **2. Alternative analyses:**
>
>
>
> Following the suggestion, we conduct additional experiments with different teacher-student pairs to evaluate the effect of sharpness-gap and CKA-gap on KD-Zero's performance (refer to the following Table). These results show that both sharpness-gap and CKA-gap contribute to the distillation results. Moreover, CKA-gap is more important for distillation gains, which can be attributed to its accurate capturing of the gap between feature representation and teacher-student. We will include these experiments in the revision to better understand these components' roles. Thanks!
>
> | Teacher                      | W-40-2 | R110  | R32x4 | VGG13 | VGG13  | W-40-2 |
> | ---------------------------- | ------ | ----- | ----- | ----- | ------ | ------ |
> | Student                      | W-16-2 | R20   | R8x4  | VGG8  | MNetV2 | SNetV1 |
> | Baseline                     | 73.26  | 69.06 | 72.5  | 70.36 | 64.60   | 70.50   |
> | KD                           | 74.92  | 70.67 | 73.33 | 72.98 | 67.37  | 74.83  |
> | KD-Zero                      | 76.42  | 72.45 | 77.85 | 75.26 | 70.42  | 77.52  |
> | KD-Zero without S-gap        | 76.06  | 72.12 | 77.36 | 74.86 | 70.15  | 77.18  |
> | KD-Zero without CKA-gap      | 76.24  | 72.17 | 77.58 | 75.08 | 70.25  | 77.36  |
> | KD-Zero without S &  CKA-gap | 75.97  | 71.95 | 77.05 | 74.82 | 69.89  | 76.92  |
>
>
>
> ---
>
> **Finally,** we  hope our response could address the concerns, and we thank the reviewer again for the helpful comments. We are glad to discuss further comments and suggestions.

---

> > ### Comment · Reviewer_WY3i · 2023-08-14
> > **Thank you for the rebuttal and additional experiments.**
> >
> > My question about Table. 4 is still not addressed in the rebuttal. I understand that in the case of ViTs distillation performs better than vanilla training. My question is about the comparison between teacher and students. In Table 4 last three columns, student performance is better than teacher by a significant margin (5-6%).

---

> > > ### Author Response · Authors · 2023-08-15
> > > **Many thanks for reply and additional responses on ViT distillation.**
> > >
> > >
> > >
> > > Dear Reviewer WY3i,
> > >
> > > We greatly appreciate your assistance in helping us gain a better understanding of the concerns. The ViT distillation experiments presented in Table 4 differ from traditional CNN distillation:
> > >
> > > 1. In traditional distillation, the teacher and student models are both CNN models that share similar visual priors and training properties. Consequently, the teacher model, with more parameters, typically possesses a better accuracy prior compared to the lightweight student model.
> > >
> > > 2. In Table 4, most ViT students possess larger model sizes (e.g., DeiT-Ti with 5 million parameters) and greater capabilities than the CNN teacher (e.g., ResNet-56 with only 0.86 million parameters). Some ViT students (e.g., PVT)  outperform the CNN model (e.g., ResNet50)  in the strong regularization setting on large-scale datasets. However, when it comes to ViT distillation on small datasets, employing CNN teachers helps address the issue of ViT models struggling to train effectively from scratch. In this context, using CNN teachers in distillation is akin to auxiliary training or providing additional regularization supervision. As a result, these ViTs demonstrate their original strong representation capability after distillation and consequently outperform the CNN teacher.
> > >
> > > 3. Similar observations have been made in some customized training methods for ViT on small datasets, such as Locality Guidance \[R1\]. ViT models trained with Locality Guidance also exhibit higher accuracy than CNN models on small datasets. We would add this discussion of these works in the revision.
> > >
> > > We sincerely hope that the above response clarifies the concerns. All these helpful suggestions will be involved in the revision. Thanks again for taking the time to review our paper!
> > >
> > > **References:**
> > >
> > > \[R1\] "Locality Guidance for Improving Vision Transformers on Tiny Datasets," ECCV 2022.

---

> > > > ### Comment · Reviewer_WY3i · 2023-08-18
> > > > **Thanks for the clarification.**
> > > >
> > > > After further clarification from authors, I will keep my original rating of 'Accept'.

---

> > > > > ### Author Response · Authors · 2023-08-18
> > > > > **Thanks for the Recognition of Our Work and Rebuttal**
> > > > >
> > > > > Thank you so much for the recognition of our responses. We are glad to your positive feedback! Thanks!
> > > > >
> > > > > Following your constructive suggestions, we will make more efforts to improve our paper further.
> > > > >
> > > > > Many thanks for your constructive comments, time and patience.

---

> ### Author Response · Authors · 2023-08-18
> **Looking forward to the reply**
>
> Dear reviewers,
>
> We sincerely appreciate your valuable feedback.
>
> As the deadline for the author-reviewer discussion phase is approaching, we would like to check if you have any other remaining concerns about our paper. If our responses have adequately addressed your concerns, we kindly hope that you can consider increasing the score.
>
> We sincerely thank you for your dedication and effort in evaluating our submission. Please do not hesitate to let us know if you need any clarification or have additional suggestions.
>
> Best Regards,
>
> Authors.

---

### Official Review · Reviewer_BLS2 · 2023-07-05

**Soundness:** 2 fair
**Presentation:** 2 fair
**Contribution:** 2 fair
**Rating:** 3
**Confidence:** 5

**Summary:**

The paper proposed to design a search space for the distillation module and apply evolution search to find the optimal combination of KD module on the pre-defined search space. The experiments are conducted are three tasks.

**Strengths:**

The motivation is good, and presentation is fair.

**Weaknesses:**

KDzero is a very straightforward method that uses the concept of neural architecture search on KD modules. The novelty is limited. As a result, I expected to see strong performance gain over baselines. Unfortunately, in the current form, the experimental results are weak.

1) only two teacher-student pairs are compared on ImageNet. I suggest the author to include more advanced architecture, such as vit.

2) did not compare with state-of-the-art detection-based KD method.
The author only compares a few methods, where most of them are not kd for det. The experiments in Table 6 are insufficient to show the "scaling knowledge transfer" ability of KDzero. I believe more advanced det-kd methods should be included, such as GID/LD/Defeat. Also different types of teacher-student pairs as well as detector types, such as anchor-free detectors, should be compared.

3）Experiments on segmentation are only conducted on a small dataset (cityscapes).
Cityscapes is a small dataset. The author should conduct experiments on ADE20K to show the effectiveness of their method.

There are some other concerns:
1) No correlation test. The author should provide a correlation test to verify if there is a correlation between proxies, such as cka-gap and sharpness-gap, with the final search results.
2）The figures are too small to see, especially figures 8 and 9.
3) The author claim that KDzero reduced the capacity gap, yet no experiments are provided. CKA value cannot reflect the teacher-student gap.
4) It is known that the design of search space is critical in NAS. The search space in KD-zeros seems to rely heavily on expert experience, and the overall search space is very similar to state-of-the-art KD methods.

**Questions:**

See weakness.

**Limitations:**

The author addressed the limitation.

---

> ### Author Rebuttal · Authors · 2023-08-07
>
> Dear Reviewer BLS2:
>
> Thanks sincerely for taking the time to review our work. We truly appreciate the valuable feedback and constructive comments. We have  addressed all the concerns in detail with point-to-point responses  below：
>
> ------
>
> **Q1:** Compare to neural architecture search.
>
> **A1:** 1. As clarified in Related Work,  KD-Zero differs significantly from NAS methods regarding its search objectives, entities, and techniques. KD-Zero optimizes transforms,  distances, and weights to enhance distillation gains, while NAS methods focus on layer stacking to achieve better efficiency.
>
> 2. Although AutoML ideas apply in different fields (e.g. NAS), our KD-Zero is the first to bring AuotML into KD by searching for distillers with significant novelty and insight  **(recognized by All other Reviewers with 2 Accept & 1 Weak  Accept ratings)**.
>
> ------
>
> **Q2:** About strong performance gain.
>
> **A2:**  There are two aspects to our response:
>
> 1. KD-Zero obtains consistent 2.5% ∼ 2.9% gains on ImageNet.  Considering that the ImageNet is a challenging benchmark for CNNs in the mobile setting with small FLOPs, these gains are actually significant improvements. i.g., MobileNetV2 was about 0.5% higher than that of  ShuffleNetV1 (SOTA of that time). Similar situations are present for other datasets and tasks.
> 2. As demonstrated in Table 3, KD-Zero enables significant extra gains with other designs, such as multi-projector, multi-layer, multi-teacher, and multi-loss.
>
> ------
>
> **Q3:** About the more advanced architecture.
>
> **A3:** Following the suggestion, we provide additional results on ViT-T in the following Table. The results show KD-Zero effectively improves ViT. We will include this in the revision. Thanks!
>
> | Teacher      | Student | Teacher | Student | KD    | DeiT | DearKD-Ti | KD-Zero |
> | ------------ | ------- | ------- | ------- | ----- | ---- | --------- | ------- |
> | RegNetY-16GF | ViT-T   | 82.9    | 72.2    | 72.35 | 74.5 | 74.8      | 76.0    |
>
> ------
>
> **Q4:** About more detection-based KDs.
>
> **A4:** Following the suggestion, we provide additional  experiments on anchor-based & anchor-free detectors in the following Table. The results show that the searched distillers outperform these detection-based KDs (e.g. GID/LD/Defeat). We will include these in the revision. Thanks!
>
> | Method              | mAP  | APS  | APM  | APL  | Method         | mAP  | APS  | APM  | APL  |
> | ------------------- | ---- | ---- | ---- | ---- | -------------- | ---- | ---- | ---- | ---- |
> | RetinaNet-Res101(T) | 38.9 | 21.0 | 42.8 | 52.4 | FCOS-Res101(T) | 40.8 | 24.2 | 44.3 | 52.4 |
> | RetinaNet-Res50(S)  | 37.4 | 20.6 | 40.7 | 49.7 | FCOS-Res50(S)  | 38.5 | 21.9 | 42.8 | 48.6 |
> | GID                 | 39.1 | 22.8 | 43.1 | 52.3 | GID            | 42.0 | 25.6 | 45.8 | 54.2 |
> | LD                  | 39.0 | 23.1 | 43.2 | 51.1 | LD             | 40.6 | 24.3 | 44.1 | 52.3 |
> | Defeat              | 39.7 | 23.4 | 43.6 | 52.9 | FGD            | 42.1 | 27.0 | 46.0 | 54.6 |
> | KD-Zero             | 39.9 | 22.9 | 43.7 | 53.6 | KD-Zero        | 42.5 | 25.2 | 46.8 | 55.1 |
>
> ------
>
> **Q5**: About large-scale Seg. Datasets.
>
> **A5:**  Following the suggestion, we perform additional experiments on the ADE20K dataset (80k 512*512) and present the mIoU(%) results in the following Table. The results indicate that KD-Zero improves segmenter by 5.09% ~4.61%. We will include these experiments in the revision. Thanks!
>
> | Teacher              | Student            | Teacher | Student | SKD[36] | IFVD[53] | CWD[48] | MGD[58] | KD-Zero(ours) |
> | -------------------- | ------------------ | ------- | ------- | ------- | -------- | ------- | ------- | ------------- |
> | PSPNet-ResNet101-R18 | PSPNet-ResNet18    | 44.39   | 30.95   | 32.37   | 32.84    | 35.09   | 35.54   | 36.04         |
> | DeepLabV3-ResNet101  | DeepLabV3-ResNet18 | 45.00   | 35.28   | 35.91   | 37.03    | 39.14   | 39.56   | 39.89         |
>
> ------
>
> **Q6**: About the correlation test and  Figures' size.
>
> **A6:**  We already make correlation test in Sec. B.2  & Table 4 in Appendix. The correlation results are: CE Loss (0.755)  Sharpness-gap (0.512 ) CKA-gap (0.682), CE Loss+Sharpness-gap+CKA- gap  (0.935), proving that Sharpness-gap & CKA-gap benefit validation. We will resize Figures 8 and 9 to be clearer in the revision.
>
> ------
>
> **Q7:** About the capacity gap.
>
> **A7**: We already provide results on the capacity gap in Figure 6 and Figure 7 of Sec. 3.3, which include training curves,  T-SNE visualization, and logits-correlation. The Grad-CAM++  visualization of Figure 1 in the Appendix demonstrates that the student model distilled by KD-Zero has a smaller capability gap with the teacher. CKA, a representations gap metric, has been recently used in  KD (e.g. DPK-ICLR2023, arxiv.org/abs/2206.06067) to analyze capacity gaps.
>
> ------
>
> **Q8**: About the search space.
>
> **A8:**  Our responses are:
>
> 1. KD-Zero‘s search space is based on our rich expert experience in KD  design and SOTA KDs, allowing researchers to apply our framework without excessive expert knowledge or trial-and-error overheads.
> 2. Our search space is organized into input knowledge OPs, transform   OPs, distance OPs, and weight OPs (calibration, temperature, and weight values), with additional advanced options in Table 1. We clearly emphasize in our ablation study that our novel and insightful search space design leads to superior results (Figure 8) and search efficiency  (Figure 9) (see Sec. 3.1 and Sec. 4.4).
> 3. Our automated method does not need to avoid current hand-designing  SOTA KDs because of the different tracks &  aims. Furthermore, the distillers searched (see Table 2) differ from the existing KDs. Our framework and findings can help further the development of hand-designed KDs.
>
> ------
>
>  **Finally,  we genuinely hope that our explanations and efforts can improve the overall evaluation of our work.** We are glad to discuss further comments and suggestions.

---

> ### Author Response · Authors · 2023-08-20
> **Sincerely appreciate considering our clarifications and increasing the score**
>
> Dear Reviewer BLS2,
>
> We sincerely appreciate your valuable feedback. As the deadline for the author-reviewer discussion phase is approaching, we would like to check if you have any other remaining concerns about our paper.
>
> In the past several months, we devote lots of effort & exploration to the KD-Zero day and night. This present work is novel and insightful building on our several years of research expertise in knowledge distillation and as recognized by most Reviewers ( 2 Accept & 1 Weak  Accept ratings). We promise the work is solid and also release our codes to benefit the research community. If our responses have adequately addressed your concerns, we kindly hope that you can consider increasing the score.
>
> In addition, we would like to emphasize some key points that have been misunderstood:
>
> 1. **About  " straightforward ":** We argue that KD-Zero to direct adopt the first automated search framework to prevent manual tuning and exploration is helpful for the development of KD applications and research.  In addition, we also carefully design and optimize our search space and search algorithms (also approved by you and other reviewers). Our well-developed KD-Zero can help researchers to find optimal distillation configurations directly in new tasks and scenarios.
> 2. **About novelty compared to NAS:** As emphasized in our rebuttal, our KD-Zero and NAS are on completely different technological routes in terms of search space, search objects, and application areas. We don't think our novelty should be weakened because we adopt AutoML ideas, which are very broadly applied in domains such as NAS and HPO (as clarified in our related work). Our work is the first to explore AutoML in KD design, which has significant benefits for the extensive knowledge distillation community.
> 3. **About performance:** our searched single-loss already outperforms all recent SOTA methods, which already proves the superiority of our method. Moreover, in our orthogonal experiments in Table 3, we also demonstrate that our KD-Zero can be further improved with mult- losses, multiple layers, multiple teachers, and advanced alignment designs (as highlighted in our rebuttal and the paper). AutoML methods including our KD-Zero and other NAS & HPO methods primarily have advantages in process automation and efficiency compared to manual design and tuning, not merely accuracy improvements.   The KD techniques have been heavily studied and optimized in different benchmarks in the past years.  Some recent SOTA methods that we compare have been well-optimized and close to the upper bound accuracy determined by the teacher's model. We believe our performance and efficiency gains are solid enough compared to these hand-designed SOTA KDs.
> 4. **About additional experiments and details:** we have added and compared different Det-KDs methods to the ADE20K experiments and clarified some analyses that were overlooked. These additional results demonstrate the superiority of our method. Moreover, as a general KD approach, our evaluations for KD-Zero in classification and common downstream tasks are already sufficient and solid, following the recent SOTA general KDs. We argue that it is understandable that our original versions did not include ADE20K experiments, as almost specially designed KDs[36,53,48] for semantic segmentation were also not tested on ADE20K. For additional details on in Q6 & Q7, please carefully consider our rebuttal and the results in the paper to clarify any misunderstandings. We will involve these in the revision following your suggestions. Thanks!
>
>
>
> We are very sad that our efforts and ideas in KD-Zero are misunderstood in your original comments. In addition, we are quite nervous about the absence of any response from you in the past week after we carefully submitted our responses.  **At present, novelty and presentation of our work have been well recognized by all other Reviewers with 2 Accept & 1 Weak Accept ratings. We sincerely hope that you will consider our clarification and improve the rating.** We sincerely thank you for your dedication and effort in evaluating our submission. Please do not hesitate to let us know if you need any clarification or have additional suggestions.
>
> Best Regards,
>
> Authors.

---

> > ### Author Response · Authors · 2023-08-21
> > **The deadline for the author-reviewer discussion phase is approaching!**
> >
> > Dear Reviewer BLS2,
> >
> > As the deadline for the author-reviewer discussion phase is approaching, we would like to check if you have any other remaining concerns about our paper. If our responses have adequately addressed your concerns, we kindly hope that you can consider increasing the score.
> >
> > We sincerely thank you for your dedication and effort in evaluating our submission. Please do not hesitate to let us know if you need any clarification or have additional suggestions.
> >
> > Best Regards,
> >
> > Authors.

---

> > > ### Author Response · Authors · 2023-08-21
> > > **Seeking your score improvement**
> > >
> > > Dear Reviewer BLS2,
> > >
> > > We hope this email finds you well. We would like to express our gratitude once again for your time and effort in reviewing our paper titled "KD-Zero: Evolving Knowledge Distiller for Any Teacher-Student Pairs." Your valuable and constructive comments have been immensely helpful in improving the quality of our work.
> > >
> > > Over the past week, we have diligently worked on addressing each concern raised during the review process. We have carefully considered your feedback and have made substantial revisions to our paper accordingly. We believe that these changes have significantly strengthened the overall contribution and clarity of our research.
> > >
> > > With that said, we eagerly await your response to our revised manuscript and would appreciate any further feedback you may have. Your input is invaluable to us and we truly value your expertise in the field.
> > >
> > > If our response and the subsequent revisions meet your expectations, we kindly request that you consider raising your rating of our paper. We believe that the enhancements we have made, based on your suggestions, have substantially improved the merits of our research and its relevance to the scientific community.
> > >
> > > Should you still have any lingering questions or concerns regarding our paper, we are more than willing to address them promptly. We are committed to ensuring that our work is of the highest standard, and we welcome the opportunity to make further improvements if necessary.
> > >
> > > Once again, we extend our heartfelt appreciation for your time, expertise, and commitment to the peer review process. We look forward to receiving your feedback and working towards the successful publication of our paper.
> > >
> > > Thank you and best regards,
> > >
> > > Best
> > >
> > > Authors

---

### Official Review · Reviewer_3PTD · 2023-07-05

**Soundness:** 3 good
**Presentation:** 4 excellent
**Contribution:** 3 good
**Rating:** 7
**Confidence:** 5

**Summary:**

This work aims to find the best distiller for different teacher-student pairs to reduce the gap caused by different capabilities and architectures. KD-Zero, the first auto-search framework for distillation was proposed, including search space design, search objectives, and search accelerate techniques. It performs state-of-the-art in multiple datasets and architectures including CNN and ViT.

**Strengths:**

- First auto-search framework for distillation.
- Rich and solid experiments, and state-of-the-art results.
- Great presentation.


**Weaknesses:**

- Efficiency, every different teacher-student pair need to search distiller, which is quite inefficient. Is there a general distiller across different teacher-student pairs?
- Reason behind the choice of the distiller. Some analysis and practical Guidance for distiller choice are given, but the conclusion is quite shallow. What is the key design for different architecture pair? What differences in the architecture pair make the distiller the best one?
- The search space designs need a broader exploration, such as a line of research on decoupling distillation into target class and non-target class[1], or knowledge distillation with label smoothing[2][3], etc.


[1]arxiv.org/abs/2203.08679

[2]arxiv.org/abs/2107.00181v1

[3]arxiv.org/abs/2207.12980

**Questions:**

The results are clear in general, and I have the following questions：
- Are there any insights into why the distiller benefits ViT so well?
- Is there a general distiller across different teacher-student pair?
- Table 5 in the supplement indicates CIFAR100 or Imagenet. What is the time cost of searching in imagenet?



**Limitations:**

yes

---

> ### Author Rebuttal · Authors · 2023-08-07
>
> Dear Reviewer 3PTD,
>
> Thank you for your thoughtful comments and questions. We sincerely thank the reviewer for the positive comments on our work! We would like to address the concerns as follows:
>
> ---
>
> **Q1:** About general distiller and efficiency.
>
> **A1:**  Yes. General distillers are available in our search.
>
> 1. KD-Zero searched distillers can generalize well to different teacher models:  In Figure 2 and Tables 1&2 in the Appendix, we actually use the distillers searched in ResNet110 (T)->ResNet20 (S) to transfer directly to the distillation of ResNet20 by different teacher models. The searched distillers improve the student model and outperform KD, DIST, and WSLD. The same findings are verified by the WRN-16-2 experiment.
> 2. KD-Zero searched distillers can generalize well to different teacher-student pairs. The results on downstream tasks demonstrate that searched distillers can work across different models and datasets (see Tables 6&7). In addition, we perform additional generalization experiments to apply the searched distillers in Table 2 with different teacher-student pairs in the following Table. The results show that the searched distillers distill well in different teacher-student pairs.
> 3. For efficiency: almost AutoML methods need additional search costs. We make great efforts to achieve at least 80 times faster search than regular search by loss-rejection protocol, search space shrinkage, and early-stopping proxy setting. Meanwhile, we searched distillers that can be transferred to different teacher-student pairs without additional searching costs.
>
> | Teacher                    | W-40-2 | R110  | R32x4 | VGG13 | W-40-2 |
> | -------------------------- | ------ | ----- | ----- | ----- | ------ |
> | Student                    | W-16-2 | R20   | R8x4  | VGG8  | SNetV1 |
> | Teacher                    | 75.61  | 74.31 | 79.42 | 74.64 | 75.61  |
> | Student                    | 73.26  | 69.06 | 72.50 | 70.36 | 70.50  |
> | KD                         | 74.92  | 70.67 | 73.33 | 72.98 | 74.83  |
> | WSLD                       | 75.30  | 71.53 | 74.79 | 74.36 | 75.09  |
> | KD-Zero                    | 76.42  | 72.45 | 77.85 | 75.26 | 77.52  |
> | KD-Zero Searched on W-16-2 | 76.42  | 71.76 | 77.65 | 74.65 | 76.85  |
> | KD-Zero Searched on R20    | 76.12  | 72.45 | 77.57 | 74.86 | 77.15  |
> | KD-Zero Searched on R8x4   | 76.24  | 71.85 | 77.85 | 74.88 | 76.76  |
> | KD-Zero Searched on VGG8   | 75.97  | 72.02 | 77.23 | 75.26 | 76.62  |
> | KD-Zero Searched on SNetV1 | 75.76  | 71.63 | 76.85 | 74.53 | 77.52  |
>
> ---
>
> **Q2:** About the key design and differences in the architecture pair make the best distiller.
>
> **A2:**  Based on searched distillers analyses, we present KD designs guidance summary in Sec. 3.3 and Supp. Sec. C1. Following the suggestion, we will add more analyses in the revision.
>
> 1. Normalized-based transforms (e.g. normC, min-max) and losses (e.g. ℓPearson, ℓCosine ), shape-based transforms (e.g. batch, scale-r1, channel) and focal-calibration, small weight factors are key designs.
>
> 2. KD design analysis from an architectural perspective:
>
>    a. Wider teacher-student models usually opt for channel-wise ops (e,g. normC in VGG8, channel in R8x4).
>
>    b. ViT models (e,g. DeiT, T2T, PVT) tend to opt for feature knowledge, which contains rich local information.
>
>    c. Large teachers (e.g., R110, WRN-40-2) prefers focal-calibration to avoid overconfidence.
>
>    d. Some cross-architecture models (e.g., SNV1, SNV2, ViT) also often shape-based transforms to better align their knowledge. This suggests that shape-based transforms help bridge heterogeneous models.
>
>    e. On large-scale ImageNet datasets, R18 adopts ℓPearson , indicating ℓPearson’s suitability for large-scale tasks.
>
> ---
>
> **Q3:** About the search space designs.
>
> **A3:** Many thanks for this really great suggestion. We agree that considering recent insightful KD designs. like decoupling distillation (e.g, DKD[1]), and KD with label smoothing (e.g., IE-KD[2] and Zipf's LS[3]) will add more depth to our KD-Zero.  During rebuttal, we explore distiller search with target class & non-target class logits as inputs and obtain promising results (see the following Table). We will incorporate these insights into our search space design and discuss them in the revision.
>
> | Teacher                                    | W-40-2 | R32x4 | VGG13 | VGG13  |
> | ------------------------------------------ | ------ | ----- | ----- | ------ |
> | Student                                    | W-16-2 | R8x4  | VGG8  | MNetV2 |
> | Baseline                                   | 73.26  | 72.50 | 70.36 | 64.60  |
> | DKD                                        | 76.24  | 76.32 | 74.68 | 69.71  |
> | KD-Zero (target class &  non-target class) | 76.69  | 77.33 | 75.19 | 70.22  |
>
> ---
>
> **Q4:** About the distiller benefits ViT so well.
>
> **A4:** Analyses are as follows and we will add these in the revision. Thanks!
>
> 1. All our searched distillers use CNN teacher's features as input knowledge. These local and hierarchical features can help ViT train from scratch on CIFAR-100.
> 2. Our distillers employ normalized-based transforms (e.g. normN) & losses (e.g. ℓPearson, ℓCosine ), shape-based transforms (e.g. batch, scale-r1, channel) , attention-based transforms (e.g. catt, mask), which can reduce CNN-ViT feature gaps.
>
> ---
>
> **Q5:** About the general distiller across different teacher-student pair.
>
> **A5:** Please see response in **A1**.
>
> ---
>
> **Q6:** About the time cost of searching in ImageNet.
>
> **A6:** The search phase on ImageNet took about 12 GPU-hours . We will add these in the revision. Thanks!
>
> ---
>
> **Finally,** we hope that our response could satisfy the reviewer's concerns, and we value your input and believe it will greatly help us improve our manuscript. We are glad to discuss further comments and suggestions.

---

> ### Author Response · Authors · 2023-08-20
> **Sincerely appreciate considering our clarifications and increasing the score**
>
> Dear Reviewer 3PTD,
>
> We sincerely appreciate your valuable feedback. As the deadline for the author-reviewer discussion phase is approaching, we would like to check if you have any other remaining concerns about our paper.
>
> In the past several months, we devote lots of effort & exploration in the KD-Zero day and night. This present work is novel and insightful building on our several years of research expertise in the field of knowledge distillation and as recognized by most Reviewers ( 2 Accept & 1 Weak  Accept ratings). We promise the work is solid and also release our codes to benefit the research community. If our responses have adequately addressed your concerns, we kindly hope that you can consider increasing the score.
>
>
> We sincerely thank you for your dedication and effort in evaluating our submission. Please do not hesitate to let us know if you need any clarification or have additional suggestions.
>
> Best Regards,
>
> Authors.

---

> > ### Comment · Reviewer_3PTD · 2023-08-20
> >
> > Dear authors,
> >
> > Thanks for your answer, and I keep my original rating of 'Accept'.

---

> > > ### Author Response · Authors · 2023-08-20
> > > **Thanks for the Recognition of Our Work and Rebuttal**
> > >
> > > Thank you so much for the recognition of our responses. We are glad to your positive feedback! Thanks!
> > >
> > > Following your constructive suggestions, we will make more efforts to improve our paper further.
> > >
> > > Many thanks for your constructive comments, time and patience.

---

### Official Review · Reviewer_AvFD · 2023-07-07

**Soundness:** 3 good
**Presentation:** 3 good
**Contribution:** 3 good
**Rating:** 6
**Confidence:** 4

**Summary:**

This paper presents KD-Zero, which utilizes evolutionary search to automatically discover promising distiller from scratch for any teacher-student architectures. The discovered distiller can address the capacity gap and cross-architecture challenges for any teacher-student pairs in the final distillation stage. The authors also  conduct extensive experiments on classification, detection, and segmentation to validate KD-zero's effectiveness.

**Strengths:**

1) The writings and organizations are good.
2) The decomposition of the generalized distiller makes sense and the authors have conducted ablations with other search algorithm choice.
3) The experiments are comprehensive including various vision tasks.

**Weaknesses:**

Though the author mentioned its efficiency as one of its advantages. I still worry about its efficiency in terms of training.
Traditional KD forms indeed needs hyper-param tuning but one expert can choose the proper temperature or other hyper-parameters according to the class number and other conditions, thus reducing the tuning laboring. The KD-Zero involves additional searching phase and I found no searching overhead reported for this phase.

**Questions:**

1) Is there efficiency comparison with other KD forms?

2) In table 4, some of the 'Auto-KD' performance even surpass the teacher model by large margin. why?

**Limitations:**

The author mentioned some limitations in the appendix.
In addition to the training efficiency, I currently find no other limitation for application.

---

> ### Author Rebuttal · Authors · 2023-08-07
>
> Dear Reviewer AvFD,
>
>  Thanks for the valuable feedback. We have tried our best to address all concerns in the last few days. If the reviewer finds our response adequate, we would really appreciate it if the reviewer considers raising the score. Please see our responses below one by one：
>
>
>
> ----
>
> **Q1:** About the  Search & Training Efficiency.
>
> **A1:**   Our responses are as follows, and we will add these in the revision. Thanks!
>
> 1. We clarify the presentation search overhead and training speed in B.4 & B.5 in the Appendix. KD-Zero requires around 11GPU-hours for search, yet searched distillers share similar training speeds with simple KD methods (e.g., KD & DIST) and are superior to complex KD methods (e.g., CRD).
> 2. Almost all AutoML methods, including KD-Zero introduce additional search costs. We highlight the efficiency is compared to the regular search, and our loss-rejection protocol and search space shrinkage improve 4 x search efficiency. Moreover, we adopt 5% early-stopping proxy setting to reduce the 20x search cost.
> 3. KD-Zero addresses new teacher-student pairs without expert knowledge by allowing automated distillers search strategies. We can summarise new expert knowledge from the automatically searched distillers (see our Practical Guidance in Section 3.3 and Section C.1in the Appendix).
> 4. Our searched distillers are transferable and general. For example, in Figure 2 in the main paper and Tables 1&2 in the Appendix, our distillers searched in one teacher can be directly transferred to different teacher distillers under both surpass some of the hand-designed methods for distillation gaps (e.g. DIST & WSLD) without additional search process. In addition, we perform additional generalization experiments to apply the searched distillers for different teacher-student pairs in the following Table. The results show that the searched distillers indeed distill well in different teacher-student pairs.
>
> | Teacher                    | W-40-2 | R110  | R32x4 | VGG13 | W-40-2 |
> | -------------------------- | ------ | ----- | ----- | ----- | ------ |
> | Student                    | W-16-2 | R20   | R8x4  | VGG8  | SNetV1 |
> | Teacher                    | 75.61  | 74.31 | 79.42 | 74.64 | 75.61  |
> | Student                    | 73.26  | 69.06 | 72.50 | 70.36 | 70.50  |
> | KD                         | 74.92  | 70.67 | 73.33 | 72.98 | 74.83  |
> | WSLD                       | 75.30  | 71.53 | 74.79 | 74.36 | 75.09  |
> | KD-Zero                    | 76.42  | 72.45 | 77.85 | 75.26 | 77.52  |
> | KD-Zero Searched on W-16-2 | 76.42  | 71.76 | 77.65 | 74.65 | 76.85  |
> | KD-Zero Searched on R20    | 76.12  | 72.45 | 77.57 | 74.86 | 77.15  |
> | KD-Zero Searched on R8x4   | 76.24  | 71.85 | 77.85 | 74.88 | 76.76  |
> | KD-Zero Searched on VGG8   | 75.97  | 72.02 | 77.23 | 75.26 | 76.62  |
> | KD-Zero Searched on SNetV1 | 75.76  | 71.63 | 76.85 | 74.53 | 77.52  |
>
> ---
>
> **Q2:** About the efficiency comparison with other KD forms.
>
> **A2:** This comparison should be discussed in search and transfer scenarios:
>
> 1. For transfer scenarios: The total overheads are close to simple KD because we can directly transfer the searched distillers without additional search costs. In B.5 in the Appendix, we present the training efficiency comparison of the searched distillers and other KD methods. Our searched distillers share similar training speeds with simple KD methods (e.g. KD & DIST) superior to complex KD methods (e.g. CRD).
> 2. For search scenarios: The overall efficiency includes the search part in this case. The additional search overheads (see Appendix Table 5) would be 3-5x than traditional KD training costs. However, these KD methods also need additional overheads for hands-on design and debugging, which can be more time-consuming than our well-optimized search process.
> 3. We will further optimize the description of search costs and add these discussions in the revision, thanks!
>
> ---
>
> **Q3:** About some results in Table 4.
>
> **A3:** Yes. ViT distilled by CNN teachers achieves superior results than teachers. The reasons are as follows:
>
> 1. ViT relies on self-attention mechanisms to capture global dependencies among image patches, allowing it to effectively model long-range interactions. Such ViT architecture performs well with large-scale datasets and strong regularization. However, small datasets are not sufficient to extract hierarchical information for ViT. This leads to weak results for these ViT models vanilla trained from scratch on CIFAR-100.
>
> 2. CNN teacher models are good at extracting local and hierarchical feature knowledge, which ViT models need exactly. With the auxiliary supervision from the CNN teacher, the ViT model can train well on CIFAR-100 and present strong performance.
>
> 3. we provide additional results on ViT-T on large-scale ImageNet datasets in the following Table. Our KD-Zero can improve the ViT from 72.0 to 76.0, surpassing the traditional KD methods e.g., DeiT by large margins, demonstrating our method's ability to generalize.
>
>    | Teacher      | Student | Teacher | Student | KD    | DeiT | DearKD-Ti | KD-Zero |
>    | ------------ | ------- | ------- | ------- | ----- | ---- | --------- | ------- |
>    | RegNetY-16GF | ViT-T   | 82.9    | 72.2    | 72.35 | 74.5 | 74.8      | 76.0    |
>
> ----
>
> **Finally,** we hope these responses address the concerns, and we appreciate the constructive feedback. We are committed to improving our manuscript and believe the insights will significantly contribute to this goal. We are glad to discuss further comments and suggestions. If the reviewer finds our response adequate, we would really appreciate it if the reviewer considers raising the score sing the score.

---

> ### Author Response · Authors · 2023-08-18
> **Looking forward to the reply**
>
> Dear reviewers,
>
> We sincerely appreciate your valuable feedback.
>
> As the deadline for the author-reviewer discussion phase is approaching, we would like to check if you have any other remaining concerns about our paper. If our responses have adequately addressed your concerns, we kindly hope that you can consider increasing the score.
>
> We sincerely thank you for your dedication and effort in evaluating our submission. Please do not hesitate to let us know if you need any clarification or have additional suggestions.
>
> Best Regards,
>
> Authors.

---

> > ### Comment · Reviewer_AvFD · 2023-08-19
> > **Reply**
> >
> > Thank you for the response. After reading it and other reviewer's comments, I keep the initial 'weak accept'.

---

> > > ### Author Response · Authors · 2023-08-19
> > > **Thanks for the Positive Feedback**
> > >
> > > Thank you so much for the recognition of our responses. We are glad to your positive feedback! Thanks!
> > >
> > > Following your constructive suggestions, we will make more efforts to improve our paper further.
> > >
> > > Many thanks for your constructive comments, time and patience.

---

### Author Rebuttal · Authors · 2023-08-10

## The Summary of Our Responses to All Official Reviews



Dear Reviewers, Area Chairs and Program Chairs,

We sincerely thank all four reviewers for their thorough and constructive comments. We are glad that the novelty, the presentation, the basic experiments, and the performance of our work have been well recognized by most Reviewers with **2 Accept & 1 Weak Accept pre-ratings**. More encouragingly, Reviewer AvFD, L7DA, and nBFP think our first automated kd search work may be insightful for the KD community.

In the past weeks, we carefully improved the experiments (using all computational resources we have), the clarifications, and the discussions of our work to address the concerns, the questions, and the requests of all four reviewers.



**Finally**, based on the constructive comments by all four reviewers and our responses, **we will carefully revise the manuscript of our work**. We hope our detailed responses and the revised manuscript are helpful to address the concerns, the questions, and the requests of all four reviewers.

---

### Author Response · Authors · 2023-08-22
**Thank all Reviewers and Area Chairs for your great efforts and insightful comments!!**

Dear Reviewers and Area Chairs,

We sincerely appreciate your great efforts, insightful comments, and the constructive suggestions you have provided once again! Through our discussions and the reviewers' responses, it appears that we have effectively addressed the major concerns raised by everyone. In particular, **three Reviewers (e.g., Reviewer AvFD, Reviewer 3PTD,  Reviewer WY3i) recognized our response and kept the positive feedback (2 Accept & 1 Weak Accept ratings).** Although Reviewer BLS2  has not replied or read our clarifications, our response clearly addresses his concerns and we believe that he will improve his rating and provide a positive final review if he reads our response. This outcome has greatly benefited us, and we would like to express our gratitude to all of you for your support!

We firmly believe that our framework KD-Zero for auto-search knowledge distiller plays a significant role in advancing the community. And we are committed to making our code and training details publicly available. Moreover, we are eager to engage in further discussions with you to enhance our understanding of the domain and further improve the quality of the paper.

And we deeply appreciate that if you could reconsider the score accordingly. We are always willing to address any of your further concerns.

Best regards,

the Authors

---

### Decision · Program_Chairs · 2023-09-21

**Decision:**

Accept (poster)

**Comment:**

This paper proposes an auto-search framework to identify the best distillation method for a given teacher-student pair. In particular, the paper explores a search space with three key components -- knowledge transformations,  distance functions, and loss weights. The paper shows promising results on various vision tasks. In addition, the paper proposed various criteria such as loss rejection protocol and search space shrinkage to reduce the overhead of the auto-search.

Most of the reviewers recognized the value of this work as it's the first work that leverages auto-search for distillation. Various questions regarding the overhead of the auto-search, transferability of obtained distillation methods to other teacher-student pairs, better ablation, and key takeaways from the paper came up during the review process. The authors did a great job at resolving all of these concerns and provided multiple additional experiments to further strengthen their contributions. One of the reviewers raised concerns about the submission while asking for comparisons with additional baselines and experiments on larger segmentation datasets. The authors have satisfactorily addressed both of these asks from the reviewer.